# AcuB senses cellular energy charge to coordinate acetyl-CoA synthesis in bacteria

Markus Janetzky [1], Norman Geist [2], Sabrina Schulze [1], Henriette Rückert [1], Kimani Gatzemeyer[1], Gottfried J. Palm [1], Leona Berndt[1], Britta Girbardt[1], Daniel Weis[1], Ina Menyes[3], Norma Welsch[4], Thomas Schweder [4], Mark Dörr [3], Stefan Kemnitz[5], Uwe T. Bornscheuer [3], Mihaela Delcea[2] & Michael Lammers [1] ✉

Bacteria adjust their metabolism to the cellular energy state. AMP-forming acetyl-CoA-synthetase AcsA generates acetyl-CoA from acetate, ATP and CoA. In *Bacilli*, including *Bacillus subtilis* and *Geobacillus stearothermophilus*, AcsA is reversely transcribed upstream of the *acu*-operon encoding for the proteins AcuA, AcuB and AcuC. Lysine-acetyltransferase AcuA uses acetyl-CoA to acetylate and inactivate AcsA, while AcuC re-activates AcsA activity by deacetylation. How the counteracting activities of AcuA and AcuC are regulated is not understood. Here, we close this gap of knowledge and perform a structure-function analyzes on AcuB. These reveal AcuB forming a scissor-shaped dimer with each monomer consisting of an N-terminal Bateman domain binding to adenine nucleotides and a C-terminal ACT domain. Structural and biochemical studies as well as molecular dynamics simulations support that AMP bound AcuB binds and inhibits AcuC. Our data describe another layer of regulation of AcsA activity in Firmicutes coordinating acetate assimilation and dissimilation by the energy sensor AcuB.

B*acillus subtilis* is ubiquitously distributed and can be isolated from soil and sea water. Moreover, it colonizes plant surfaces and the animal gut, including the human intestine[1]. In order to survive, bacteria have to be able to respond to changing environmental conditions and adapt to stress situations such as limited nutrient availability. This includes the ability to use alternative sources of nitrogen and carbon[2–6]. To react on these changing conditions and to optimally use nutrients *B. subtilis* has developed sophisticated regulatory systems enabling them to sense the nutrient composition and to adjust gene expression programs accordingly[7–9].

The master transcriptional regulator catabolite control protein A (CcpA) binds to catabolite responsive elements (*cre*) in operator sequences often located within the promoter regions of genes or

within the coding regions thereby directing carbon flow or stress tolerance[10–14]. CcpA belongs to the LacI-GalR-family and was shown to activate transcription of some genes including the gene encoding for acetate kinase (Ack) involved in acetate dissimilation, i.e., generation of acetate from acetyl-CoA by phosphotransacetylase (Pta) and Ack[11,15–17]. However, more often CcpA was shown to act in concert with phosphorylated histidine phosphocarrier protein (P-HPr), a component of the phosphotransferase sugar uptake system (PTS). This results in repression of genes for utilizing secondary carbon sources, including acetate, when preferred primary carbon sources such as glucose are available[4,11,15,18–21]. This transcriptional regulation is known as glucose repression or carbon catabolite repression (CCR)[2,9,22]. Overall, CcpA was reported being a negative regulator for genes of

[1]Department of Synthetic and Structural Biochemistry, University of Greifswald, Institute of Biochemistry, Greifswald, Germany. [2]Department of Biophysical Chemistry, University of Greifswald, Institute of Biochemistry, Greifswald, Germany. [3]Department of Biotechnology & Enzyme Catalysis, University of Greifswald, Institute of Biochemistry, Greifswald, Germany. [4]Department of Pharmaceutical Biotechnology, University of Greifswald, Institute of Pharmacy, Greifswald, Germany. [5]Department for High Performance Computing, University of Greifswald, University Computing Center, Greifswald, Germany. ✉e-mail: michael.lammers@uni-greifswald.de

carbon utilization and a positive regulator for genes involved in production and secretion of acetate[12,23,24]. An operon under the transcriptional control of CcpA is the *acu*-operon in *B. subtilis*[12,23,25]. Notably, CcpA is encoded downstream of the *acu*-operon separated by two open reading frames encoding for motility proteins MotS and MotP[26]. CcpA was reported to repress *acu*-operon expression under conditions of high glucose availability by binding to *cre*-sites in the promoter region[23].

The *acu*-operon is reversely transcribed and located downstream of the *ascA*-gene encoding for AMP-forming acetyl-CoA synthetase (AcsA), which is also under the control of CcpA[26]. The name *acu* originates from acetoin utilization as the *acu*-operon was originally regarded to be involved in breakdown acetoin and 2,3-butanediol[26]. Later studies showed that the gene products are involved in regulation of AcsA activity by lysine acetylation and deacetylation. The *acu*-operon consists of the genes *acuA*, *acuB* and *acuC*. The gene *acuA* encodes for the lysine acetyltransferase AcuA belonging to the GNAT-family and *acuC* encoding the classical $Zn^{2+}$-dependent lysine deacetylase AcuC[27,28]. The role of AcuB encoded by *acuB* is totally unresolved and was investigated in this study. Our laboratory and others recently uncovered the molecular mechanisms of regulation of AcsA activity by acetylation and deacetylation catalyzed by AcuA and AcuC, respectively[29–31]. In brief, AcuA acetylates AcsA at a lysine side chain in the C-terminal region, i.e., K549 in *B. subtilis* AcsA, which results in inactivation of AcsA activity as it affects both half-reactions, i.e., the adenylation reaction and the thioester forming reaction, to generate acetyl-CoA from acetate, ATP and CoA[32,33]. AcuA forms a stable complex with AcsA, which is resolved by acetylation of AcsA[31]. The dissociated AcsA can subsequently be re-activated by deacetylation catalyzed by AcuC[29,31].

Acetate dissimilation is observed under conditions of carbon overflow including high cellular acetyl-CoA concentrations. This results in production and excretion of acetate, followed by a decrease in pH of the extracellular space and as a consequence accumulation of cellular acetate up to millimolar concentrations[15]. Under these conditions, Ack and Pta can also convert acetate back to acetyl-CoA (acetate assimilation). Moreover, acetate assimilation, i.e., generation of acetyl-CoA from acetate, is also catalyzed by AcsA and occurs under conditions of low cellular concentrations of glucose[15]. It has been suggested that these assimilative and dissimilative pathways are in part coordinated by intrinsic characteristics of the enzymes involved, such as $K_M$-values for acetate, and the intracellular concentrations of acetate[33]. Along that line, studies performed on *Escherichia coli* AcsA showed that wild type cells can grow on wide concentrations (2.5 to 50 mM) of acetate. However, upon genomic deletion of *acs* cells grow poorly on low concentrations, while those deleted for *ack* and *pta* grow poorly on high concentrations[33]. Deletion of *acs*, *ackA*, and *pta* resulted in a strain unable to grow on acetate at any concentration tested[33]. The reaction catalyzed by AcsA is irreversible due to the release of pyrophosphate in the first half reaction, i.e., activation of acetate by adenylation to build acetyl-AMP. Moreover, the $K_M$-value of AcsA for the substrate acetate was reported to be in the micromolar range[15,34].

In contrast, the equilibrium of reactions catalyzed by Pta and Ack are strongly on the side of acetate dissimilation, i.e., generation of acetate from acetyl-CoA, as the $K_M$-value of Ack for acetate is in the millimolar range[35]. In turn, this shows that the back reaction, i.e., production of acetyl-CoA by Ack and Pta, occurs only under high cellular acetate concentrations. It is not understood how AcsA activity is switched-off under high cellular acetate concentrations to prevent two active parallel pathways resulting in assimilation of acetate to produce acetyl-CoA. Moreover, the activity of AcsA is precisely regulated at several layers. The first layer exists at the transcriptional level by the regulation of expression of the *acu*-operon by the transcriptional regulator CcpA sensing cellular glucose levels resulting in repressing *acu*-expression under high glucose concentrations[25]. In

turn, no regulation of AcsA-activity by lysine acetylation/deacetylation occurs confirming AcsA playing a role for acetyl-CoA production under low levels of glucose and only when secondary carbon sources such as acetate are available. Only in this case, another level of regulation of AcsA activity is realised, which is the post-translational level, by acetylation of AcsA catalyzed by AcuA and deacetylation catalyzed by AcuC[28,29,36]. Notably, *B. subtilis* encodes for an $NAD^+$-dependent sirtuin deacetylase, i.e., SrtN, which is also able to deacetylate AcsA in a $NAD^+$-dependent manner[27]. This regulation is also reported for other bacterial species and eukaryotes including humans[27,31,37–41]. The activity of AcuA competent to acetylate and inactivate AcsA depends on the availability of acetyl-CoA, or acetyl-phosphate and CoA, and on the cellular acetyl-CoA/CoA ratio, shown to modulate the activity of GNAT acetyltransferases[29].

Overall, these considerations leave two main questions unanswered. Firstly, it is unclear how the activities of the acetyltransferase AcuA and the deacetylase AcuC are coordinated apart from the availability of the acetyl-group donor acetyl-CoA. This is essential as the acetylation of AcsA by AcuA would be counteracted by the AcsA-deacetylase AcuC since both enzymes are simultaneously expressed and do not underlie a known individual regulation mechanism at the transcriptional level. We describe here that AcuB is capable to bind and inhibit AcuC in presence of AMP to coordinate AcuA and AcuC activity. Secondly, it is still an unresolved question why acetate assimilation catalyzed by AcsA does not occur under conditions of high cellular concentrations of acetate under which AcsA should be saturated assuming a micromolar $K_M$ for acetate. Again, this must be precisely coordinated as otherwise conversion of acetyl-CoA by Pta/Ack to produce acetate (acetate dissimilation) would always outcompete AcsA activity producing acetyl-CoA from acetate (acetate assimilation). In addition, under high cellular acetate concentrations in the millimolar range two parallel pathways would be active to convert acetate to acetyl-CoA (acetate assimilation). It was reported earlier that SrtN and AcuC are needed in *B. subtilis* for growth under conditions of low acetate[27].

Here, we show that AcuB allows to coordinate the activity of AcsA under conditions of high cellular acetate ensuring not two active pathways generating acetyl-CoA from acetate. Notably, we also observe binding of AcuB to the *B. subtilis* stress alarmone diadenosine-tetraphosphate (Ap4A) suggesting that the activity of AcsA is thereby also adjusted under stress conditions[42–44]. We describe the role of AcuB acting as energy sensor that coordinates the activity of the AcsA-deacetylase AcuC and consequently also the activity of AcsA depending on the cellular energy state. With that, we uncover the missing link in regulating AcsA activity and the coordination of acetate assimilation and dissimilation by a precise regulation of the AcsA-acetylation and -deacetylation states driven by the energy sensor AcuB. Moreover, we provide evidence that this system is also responsive to cellular stress conditions.

## Results

### AcuB encoded in the *acu*-operon binds adenine nucleotides

The *B. subtilis acu*-operon encodes for the proteins AcuA, AcuB, and AcuC. AcuA and AcuC are acting as acetyltransferases and deacetylases for AMP-forming acetyl-CoA-synthetase AcsA (Fig. 1a). Acetylation of AcsA at K549, a C-terminal lysine side chain, was shown to result in its inactivation[29,31]. The role of AcuB is unknown. To decipher the role of AcuB we recombinantly expressed *B. subtilis* AcuB (BsAcuB) in *E. coli* BL21 (DE3) and purified the protein by Ni-NTA chromatography via its C-terminal hexahistidine tag (His_6-tag) (Supplementary Fig. 1a). The expression and purification of BsAcuB was challenging. As the protein precipitated during elution with imidazole, we eluted the protein with EDTA. The protein was stable and could be concentrated up to 20 mg/mL. However, we realised the protein precipitated upon further purification by size-exclusion chromatography (SEC). Analytical SEC

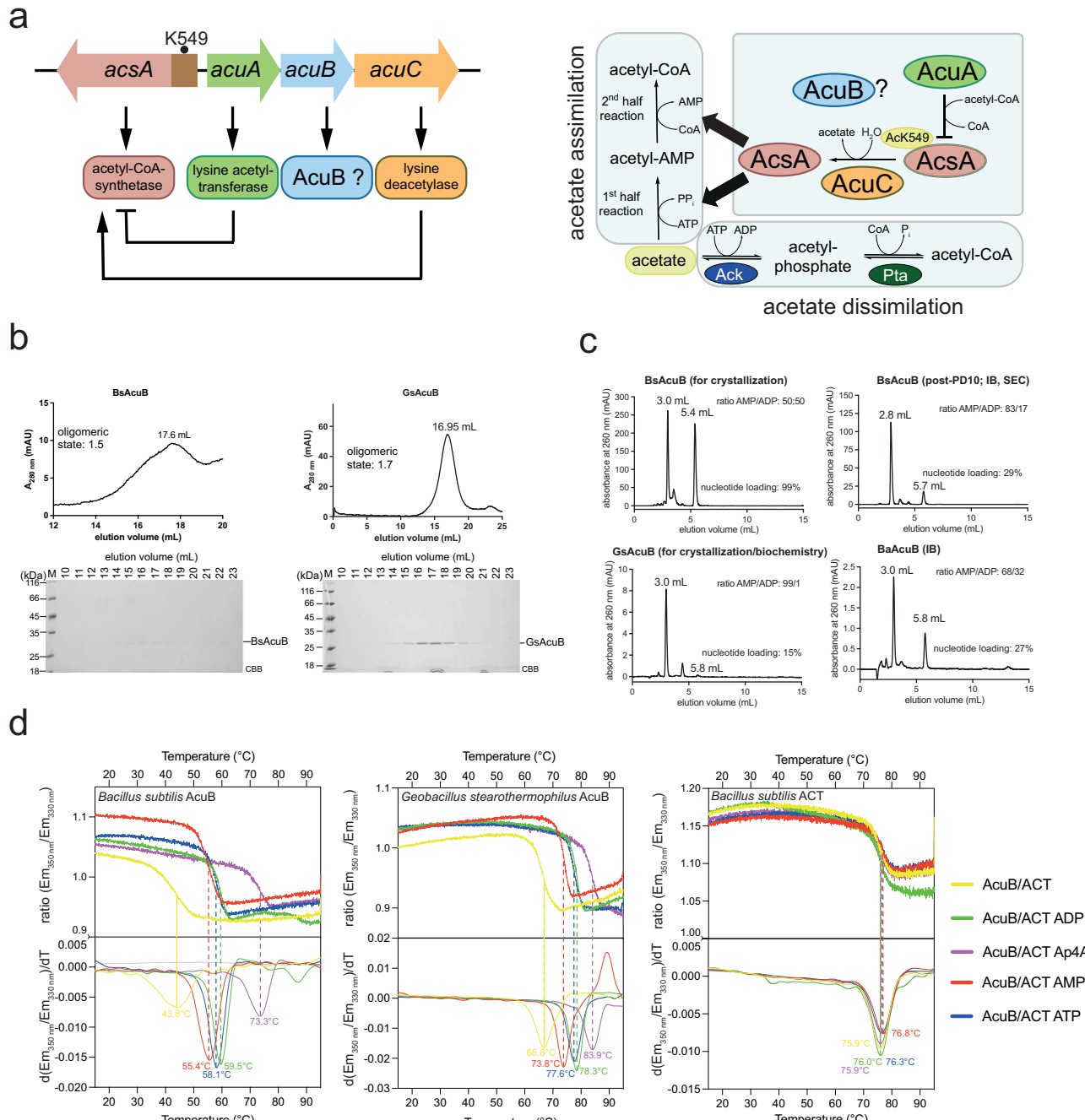

**Fig. 1 | AcuB encoded in the *acu*-operon is an adenine nucleotide binding protein. a** The *acu*-operon is distributed in Firmicutes and encodes for enzymes regulating the AMP-forming acetyl-CoA synthetase (AcsA). The lysine acetyl-transferase AcuA acetylates and inactivates AcsA, while the classical deacetylase AcuC reactivates AcsA by deacetylation. The function of AcuB is studied here. The gene *acsA* is reversely transcribed upstream of the *acu*-operon. AcsA catalyzes the production of acetyl-CoA from acetate, ATP and CoA in two half-reactions (acetate assimilation). Moreover, acetyl-CoA is converted to acetate by phospho-transacetylase (Pta) and acetate kinase (Ack) (acetate dissimilation). The figure was adapted from[29]. **b** Analytical SEC of BsAcuB and GsAcuB shows that these form dimers in solution. The elution was followed by the absorption at 280 nm ($A_{280\ nm}$) in mAU (milli absorbance units). SDS-PAGE gels show fractions of the SEC elution, which were stained by Coomassie-brilliant blue (CBB). The experiment was performed in two independent replicates ($n = 2$), one example is shown. Source data are provided as Source Data file. **c** HPLC of BsAcuB, GsAcuB and BaAcuB shows

binding to adenine nucleotides. The peak at app. 3.0 mL corresponds to AMP, the peak at app. 5.8 mL to ADP. The elution of the adenine nucleotides was followed by detection of the absorption at 260 nm ($A_{260\ nm}$). The ratio between the nucleotides is indicated together with total nucleotide-loading. The experiment was performed in two independent replicates ($n = 2$), one example is shown. Source data are provided as Source Data file. **d** Nano-differential scanning fluorimetry (nanoDSF) for BsAcuB, GsAcuB and BaAcuB was conducted in presence of adenine nucleotides. The presence of nucleotides stabilizes all proteins as shown by the shift in $T_m$ towards higher temperatures for all proteins. The upper panels show the development of the ratios of the fluorescence emission at 350 nm to 330 nm (ratio ($Em_{350\ nm}/Em_{330\ nm}$)) with increasing temperature, while the lower panels show the development of the first derivatives (d(ratio ($Em_{350\ nm}/Em_{330\ nm}$)/dT) thereof. The peak corresponds to the $T_m$ (°C) value. The experiment was performed in three independent replicates ($n = 3$), one example is shown. Source data are provided as Supplementary Data files (Supplementary Data 1-3).

showed that BsAcuB forms a dimer in solution (Supplementary Fig. 1b; Supplementary Table 1). Notably, BsAcuB was almost not-detectable at 280 nm due to its low intrinsic molar extinction coefficient (ε: 8480 M$^{-1}$cm$^{-1}$) explaining the low absorption value at 280 nm (A$_{280 nm}$). However, analysing the elution fractions by SDS-PAGE confirmed the presence of a peak with a maximum at 17.6 mL (Fig. 1b; Supplementary Fig. 1b).

AlphaFold3 structure predictions on BsAcuB suggests that it contains an N-terminal Bateman domain consisting of two cystathionine-β-synthase (CBS) modules and a C-terminal aspartate kinase, chorismate mutase, TyrA (ACT) domain (Supplementary Fig. 2). As Bateman domains are reported adenine nucleotide binding domains, we performed HPLC-analyzes and revealed that *B. subtilis* AcuB was loaded almost exclusively with AMP and ADP (Fig. 1c; Supplementary Fig. 3). For one preparation we also observed loading of BsAcuB with ATP showing that in principle BsAcuB is capable to bind ATP (Supplementary Fig. 3). Different preparations of *B. subtilis* AcuB reveal differences in the stoichiometry and the relative ratio of AMP- and ADP-loading ranging from almost 100% nucleotide loading with almost equal ratio of AMP to ADP to only 29% loading (Fig. 1c; Supplementary Fig. 3). Upon running BsAcuB on an analytical SEC or a PD10-column we lost a substantial amount of BsAcuB protein by aggregation and the AcuB protein left in solution was predominantly loaded with AMP (Fig. 1c; Supplementary Fig. 3; post-PD10). As we faced problems with protein stability and a reproducible nucleotide loading state of BsAcuB we expressed and purified also AcuB proteins from *B. anthracis* (BaAcuB) and from the thermophilic species *Geobacillus stearothermophilus* (GsAcuB) (Supplementary Fig. 1a). These proteins show a sequence identity of about 50% (BaAcuB to BsAcuB: 49%; BaAcuB-GsAcuB: 52%; BsAcuB-GsAcuB: 53%) (Supplementary Fig. 4; Supplementary Fig. 5).

Compared to BsAcuB, GsAcuB was more stable, did not precipitate and enabled analytical SEC experiments showing GsAcuB elutes as dimer (concentration: GsAcuB 10 mg/mL; BaAcuB: 8 mg/mL; Fig. 1b; Supplementary Fig. 4). HPLC-analyzes showed GsAcuB and BaAcuB were also loaded with adenine nucleotides upon expression in *E. coli* and subsequent purification (Fig. 1c; Supplementary Fig. 3). However, while BaAcuB showed similar loading as BsAcuB, with 29% total adenine nucleotide loading with almost equal amounts of AMP and ADP, GsAcuB was only loaded up to 15–23% almost exclusively with AMP (Fig. 1c; Supplementary Fig. 3). As the AcuB proteins were co-purified with AMP, ADP and to a minor proportion with ATP this suggests a moderate binding affinity in the low to medium micromolar range. We were not able to quantify the binding affinity by isothermal titration calorimetry.

To confirm the binding of adenine nucleotides to BsAcuB and GsAcuB and to further assess the impact of nucleotide binding on the stability of the AcuB proteins, we performed nano-differential scanning fluorimetry (nanoDSF)-experiments (Fig. 1d). As expected, GsAcuB was more stable compared to BsAcuB showing a melting temperature, T$_m$, of 68 °C, while BsAcuB revealed a T$_m$ of 44 °C (Fig. 1d; Supplementary Table 2). Addition of an app. 25-fold molar excess of the nucleotides AMP, ADP and ATP to both, BsAcuB and GsAcuB, resulted in a strong increase in T$_m$ of about 7 °C to 15 °C suggesting that the presence of nucleotides stabilizes the AcuB proteins to similar extent. We observed the tendency that addition of AMP has a slightly lower impact on AcuB stability compared to ADP and ATP with the latter two stabilizing both proteins similarly (Fig. 1d; Supplementary Table 2). We also assessed the effect of the stress molecule diadenosine-tetraphosphate (Ap4A) on AcuB stability and discovered it provides the strongest stabilisation for both AcuB proteins increasing T$_m$ by about 29 °C for BsAcuB and by 17 °C for GsAcuB (Fig. 1c; Supplementary Table 2). The fact that AcuB is encoded in the *acu*-operon together with the AcsA-acetyltransferase AcuA and the AcsA-deacetylase AcuC suggests that AcuB modulates AcuA and/or AcuC

activity. We reported earlier that AcuB does not affect the activity of AcuA[29]. Moreover, analytical SEC experiments performed in this study suggest that the two proteins do not form stable complexes as they do not co-elute from the column (Supplementary Fig. 1b). To this end, we next analyzed whether AcuB interferes with AcuC activity and whether the presence of different adenine nucleotides affects the interplay of AcuB and AcuC.

## AcuB inhibits AcuC deacetylase activity dependent on the nucleotide-loading state

We next questioned how the presence of AcuB affects the activity of the deacetylase AcuC and whether the presence of different adenine nucleotides modulates this interaction. For this, we used acetylated *B. subtilis* AcsA, BsAcsA AcK549, as a substrate for BsAcuC and analyzed the acetylation state of BsAcsA by immunoblotting using a specific anti-acetyl-lysine antibody as described earlier[29]. In this assay we applied the BsAcuB protein with the lowest nucleotide loading we could achieve, i.e., BsAcuB loaded to 29% with AMP and ADP (ratio 99:1) as judged by HPLC (Fig. 1c; Supplementary Fig. 3). This should enable to assess the influence of exogenously added adenine nucleotides on their capacity to modulate BsAcuC activity. Notably, we were not able to prepare full nucleotide free BsAcuB due to its tendency to precipitate without bound nucleotide. Addition of BsAcuB to BsAcuC resulted in strong inhibition of BsAcuC activity. However, this was independent from the addition of adenine nucleotides since the signal obtained for acetylated BsAcsA was unaltered compared to the control without BsAcuC (Fig. 2; Supplementary Fig. 6). This might indicate either the remaining pool of AMP/ADP bound to BsAcuB is sufficient to completely inhibit AcuC or also nucleotide free BsAcuB or only partially nucleotide loaded BsAcuB is able to bind and inhibit AcuC.

Our earlier studies on classical bacterial deacetylases suggested that recombinant BsAcuC has moderate catalytic activity compared to other enzymes[30]. As indicated, it was experimentally very challenging to work with BsAcuB due to its inherent instability, due to the difficulty to reproducibly prepare BsAcuB protein with a defined nucleotide-loading state and due to the fact that we could not prepare nucleotide free BsAcuB in order to bring the protein in a defined nucleotide loading states. Thus, we examined the more stable GsAcuB for comparison, which is to 85% nucleotide free following expression in *E. coli* and purification (Fig. 1c; Supplementary Fig. 3). While GsAcuB strongly inhibited BsAcuC under these conditions, GsAcuB was not able to inhibit GsAcuC or BaAcuC even with twentyfold molar access when no additional adenine nucleotides were added (Fig. 2; Supplementary Fig. 8). However, addition of AMP resulted in a GsAcuB protein that was competent to almost completely inhibit GsAcuC, BaAcuC and BsAcuC (Fig. 2).

Upon addition of ADP and ATP the inhibitory capacity of GsAcuB towards GsAcuC and BsAcuC was strongly impaired (Fig. 2). Notably, while GsAcuB was almost completely incompetent to inhibit GsAcuC under these conditions, BsAcuC was inhibited indicating either BsAcuC has only moderate activity or only a small fraction within the preparation is catalytically active for which the proportion of GsAcuB loaded with AMP is still sufficient to inhibit it. This might explain why we did observe inhibition of BsAcuC by BsAcuB to some extent for all conditions tested and independent from the type of nucleotide added (Fig. 2).

We confirmed these data by a Fluor-de-Lys assay using a fluorescently labeled Boc-Lys(Ac)-AMC deacetylase substrate as reported earlier (Fig. 3a)[30]. These data showed addition of AMP to GsAcuB, loaded to 23% with mostly AMP, is sufficient to inhibit the catalytic activity of GsAcuC while BsAcuB, loaded to 29% with equal amounts of AMP and ADP, does inhibit BsAcuC independent of the nucleotide added (Fig. 3a). Notably, we observed for GsAcuB that addition of ADP or ATP to BsAcuB and BsAcuC resulted in a slight statistically significant increase of BsAcuC activity (Fig. 3a). This can be explained by

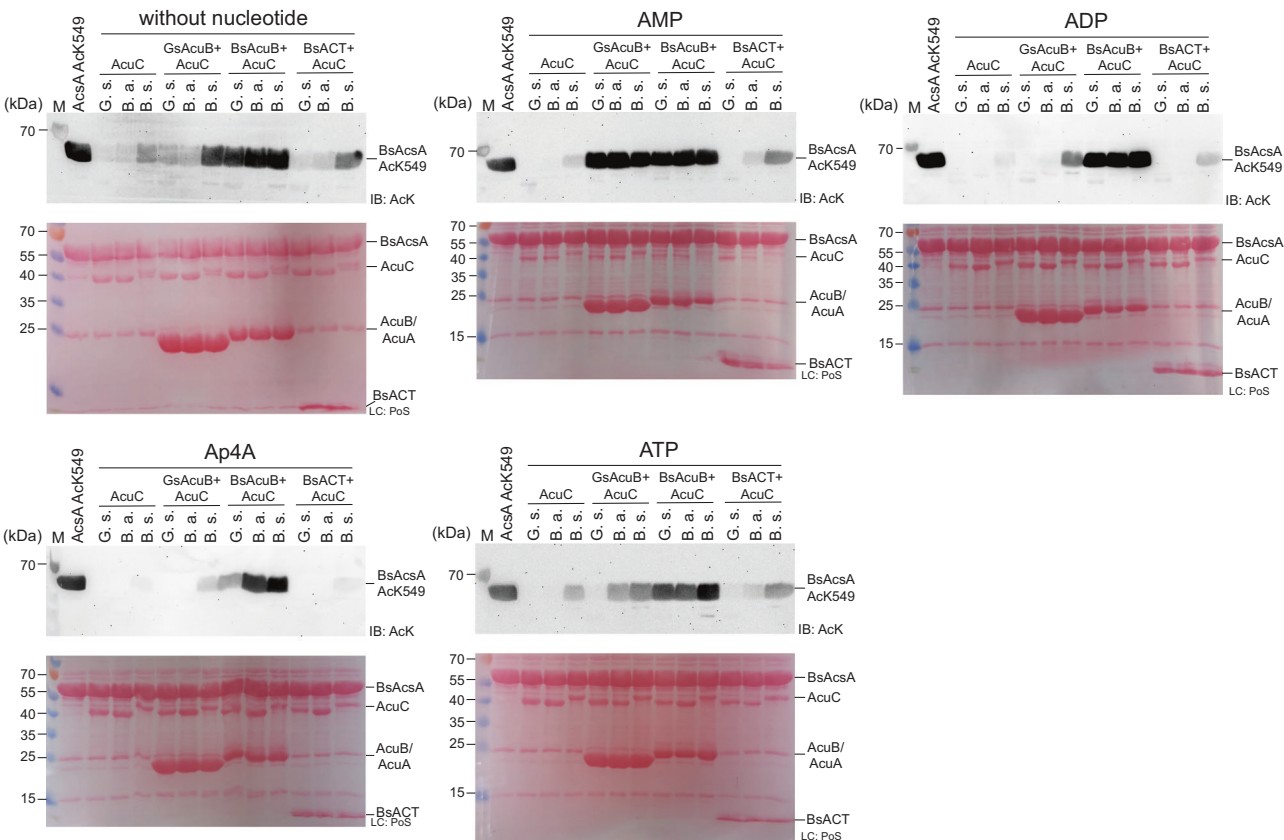

**Fig. 2 | AcuB inhibits deacetylation of AcsA catalyzed by AcuC in presence of AMP.** The acetylation state of BsAcsA (BsAcsA AcK549) was used as readout to detect the impact of different AcuB proteins, i.e., GsAcuB, BsAcuB, BaAcuB, and of the isolated *B. subtilis* ACT domain (BsACT) on its deacetylation catalyzed by different AcuC proteins, i.e., GsAcuC, BsAcuC, BaAcuC. The influence of adding different nucleotides, i.e., AMP, ADP, ATP, and Ap4A as well as a sample without addition of nucleotide was assessed. Acetylated AcsA was stained with anti-acetyl-L-lysine antibody (IB: AcK), loading control was done by staining the membrane with Ponceau S red solution (LC: PoS). The result was obtained in three independent experiments (*n* = 3), one example is shown. The quantifications and statistical analyzes are shown in Supplementary Fig. 6. Source data are provided as Source Data file.

the fact that the exogenously added ADP or ATP displaces the AMP bound to GsAcuB thereby adopting a state incompatible with binding to and inhibiting BsAcuC (Fig. 3a). We also observe that addition of the isolated BsACT domain can inhibit both BsAcuC and GsAcuC (Fig. 3a). As expected, the inhibition by the isolated BsACT domain is independent from addition of nucleotides. Moreover, the inhibition by BsACT is less efficient compared to full length AcuB proteins suggesting that AcuC interacts with both, the Bateman domains and the ACT domains present in AcuB as proposed by the AlphaFold3 model (Fig. 3a; Supplementary Fig. 2). Supporting these findings, the binding affinity of the isolated BsACT domain is lower as compared to full length BsAcuB/GsAcuB as no co-elution was observed by analytical SEC (Supplementary Fig. 8).

To further support the impact of AMP on the interaction of AcuB and AcuC we next performed analytical SEC assessing complex formation of BsAcuB/GsAcuB and BsAcuC/GsAcuC, respectively, depending on the addition of exogenous adenine nucleotides (Fig. 3b; Supplementary Fig. 9; Supplementary Fig. 10). The BsAcuB-batch analyzed in these experiments was loaded to 29% predominantly with AMP as judged by HPLC (Fig. 1c, Supplementary Fig. 3). BsAcuB alone eluted as dimer from the SEC column while BsAcuC eluted as monomer (Fig. 1b; Supplementary Fig. 1b; Supplementary Fig. 9). Incubating a twofold molar excess of BsAcuB (200 μM) with BsAcuC (100 μM) and subsequent analysis by analytical SEC reveals a small portion of BsAcuB formed a complex with BsAcuC when no exogenous nucleotide was added (Fig. 3b; Supplementary Fig. 9). As stated above, this can be explained by the part of BsAcuB being loaded with AMP after

expression in *E. coli* and purification. The elution volume of 15.2 mL corresponds to either a 2:1 BsAcuC-BsAcuB complex or a 2:2 BsAcuC-BsAcuB complex (Fig. 3b; Supplementary Table 1; Supplementary Fig. 9). The addition of exogenous AMP before incubating BsAcuB with BsAcuC strongly improved complex formation (exp. MW: 139354 Da; elution volume: 14.8 mL; calc. MW: 132553 Da; stoichiometry of complex: 2:2) while addition of neither ADP, ATP nor Ap4A did grossly affected complex formation compared to the sample without addition of nucleotides (Fig. 3b; Supplementary Fig. 9; Supplementary Table 1). For GsAcuB and GsAcuC we observe quantitative complex formation upon addition of AMP by analytical SEC (Fig. 3b; Supplementary Fig. 10; Supplementary Table 3).

In addition, we analyzed complex formation of GsAcuB and GsAcuC by nanoDSF measurements. These data suggest that AcuC alone does not bind to any of the adenine nucleotides analyzed (Supplementary Fig. 11). Moreover, addition of all nucleotides results in stabilisation of the GsAcuB protein compared to the sample without addition of nucleotides. However, all nucleotides vary in their stabilizing effect (Fig. 1d; Supplementary Fig. 11). Notably, also nanoDSF data of GsAcuB, GsAcuC and their mixture show, that in contrast to the sample without addition of exogenous nucleotides, for all samples with nucleotides the fluorescence of the mixture is not a linear combination of its components. This proofs interaction of GsAcuB with GsAcuC, most likely complex formation occurs upon addition of all adenine nucleotides to some extend (Supplementary Fig. 11). Overall, these data suggest that binding of AMP to AcuB stabilizes a conformation, which is most competent to bind to and inhibit AcuC-

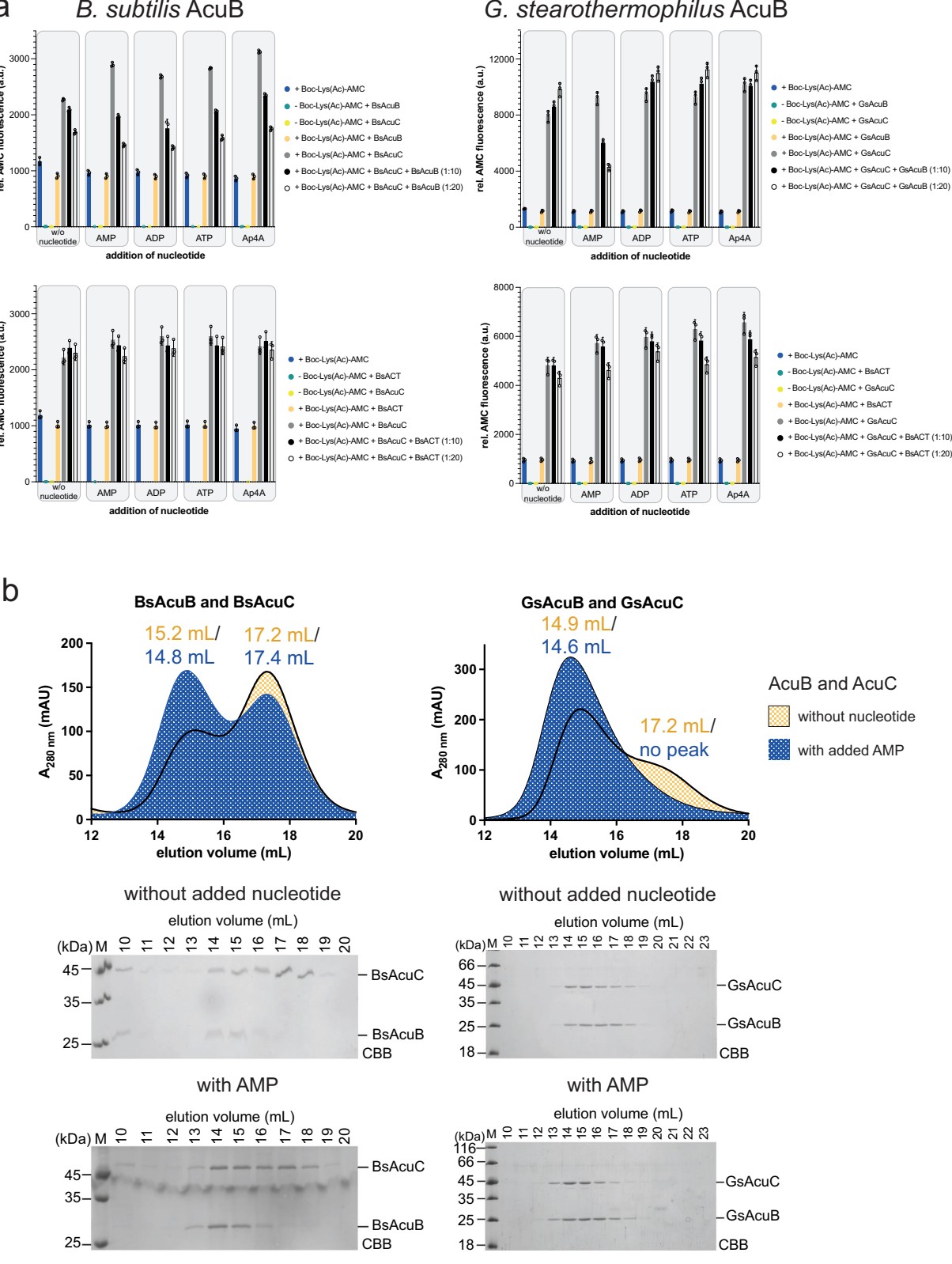

deacetylase activity, ultimately translating into a sustained inactivation of AcsA activity under conditions of low intracellular energy charge. To understand the molecular mechanisms underlying the observed impact of AcuB nucleotide binding on complex formation between AcuB and AcuC and inhibition of AcuC activity, we next analyzed AcuB by X-ray crystallography in the apo state and in complexes with various adenine nucleotides and combinations thereof.

## AcuB binds adenine nucleotides in the Bateman domains with four binding sites per dimer

We conducted structural analyzes by X-ray crystallography to understand how AcuB binds adenine nucleotides and how binding of different nucleotides only varying in the number of phosphate-groups can explain the observed differences of AcuB's capacity to bind and to inhibit AcuC. Initially, we performed crystallization approaches with a

**Fig. 3 | Addition of AMP to AcuB increases its capacity to inhibit AcuC activity and increases complex formation. a** Addition of AMP allows AcuB to inhibit AcuC activity. The impact of exogenously supplied adenine nucleotides on the capacity of full length AcuB (upper panels), i.e., BsAcuB and GsAcuB, and of the isolated BsACT domain to inhibit BsAcuC and/or GsAcuC was analyzed in a Fluor-de-Lys fluorescence assay with Boc-Lys(Ac)-AMC as substrate. For BsAcuB the addition of all tested nucleotides and no supply of exogenous nucleotide resulted in a similar capacity of AcuB to inhibit AcuC activity. Inhibition increased in presence of higher concentrations of AcuB, i.e., 1:10 AcuC:AcuB versus 1:20 AcuC:AcuB. For GsAcuB addition of AMP enabled inhibition of GsAcuC while ADP, ATP and Ap4A resulted in a significant increase in GsAcuC activity with increasing concentrations of GsAcuB. BsACT was able to inhibit both, BsAcuC and GsAcuC, with lower efficiency compared to full length AcuB. The relative fluorescence was determined and is shown in arbitrary units (a.u.). The experiment was performed in three independent replicates ($n = 3$), one example is shown. Data are presented as mean ± SD. The statistical

analyzes were performed conducting an ordinary ANOVA test ($p < 0.05$: *; $p < 0.01$: **; $p < 0.001$: ***; $p < 0.0001$: ****). Source data are provided as Source Data file. **b** Addition of AMP to AcuB and AcuC increases complex formation as shown by analytical SEC. BsAcuB and BsAcuC as well as GsAcuB and GsAcuC show an increased complex formation in presence of exogenously supplied AMP (blue elution profile) compared to the sample without addition of exogenous nucleotide (yellow elution profile). AcuB$_2$•2AcuC complex formation was also observed in the samples without addition of nucleotides due to the fraction of AcuB pre-loaded with AMP. For GsAcuB/GsAcuC addition of AMP almost quantitatively resulted in complex formation. The elution was followed by recording the absorption at 280 nm ($A_{280\,nm}$) in mAU (milli absorbance units). The SDS-PAGE gels show fractions of the SEC elution as indicated. The gel was stained by Coomassie-brilliant blue (CBB). The SEC experiment was performed in two independent replicates ($n = 2$), one example is shown. Source data are provided as Source Data file.

batch of purified BsAcuB loaded to 99% with equal amounts of AMP and ADP (ratio 50:50) (Fig. 1c; Supplementary Fig. 3). The obtained crystals diffracted up to a resolution of 2.64 Å and belonged to orthorhombic space group P2$_1$22$_1$ with three molecules of BsAcuB per asymmetric unit (Supplementary Figs. 12–14; Supplementary Table 4). Two of those monomers form an asymmetric dimer related by non-crystallographic symmetry (NCS; chain A and chain B) while the third monomer is part of a crystallographic dimer (chain C) directly lying on a crystallographic twofold symmetry axis (PDB: 9SAV; Supplementary Fig. 12). Each monomer consists of an N-terminal Bateman domain each composed of two CBS modules (Fig. 4a; Supplementary Fig. 12). Each Bateman domain binds to one AMP and one ADP molecule (Supplementary Figs. 12–14). While the archetypical CBS module fold consists of a three-stranded β-sheet with two α-helices flanking one site of the β-sheet, in BsAcuB the β-sheet is only two-stranded and forms an antiparallel β-hairpin as described also for other CBS modules (Fig. 4a). The β-hairpins of each CBS module within a Bateman domain are flanked by flexible loop regions originating from the BsAcuB N-terminus and from the flexible linker connecting the two CBS modules. Two CBS modules within each monomer associate through stacking of their β-sheets atop each other (Fig. 4a). The overall topology of each CBS module in BsAcuB is α0-α1-β1-β2-α2. The N-terminal Bateman domain of each monomer leads to the C-terminal ACT domain connected by a linker originating from α2 of the C-terminal CBS module. The archetypal ACT domain has the topology β1-α1-β2-β3-α2-β4. The BsAcuB ACT domain lacks the C-terminal β-strand resulting in a slightly modified architecture containing an antiparallel β-sheet, which is flanked by three α-helices facing the side opposite from the Bateman domain forming a β1-α1-β2-β3-α2 topology.

Overall, two BsAcuB monomers form a scissor-shaped dimer with a twofold rotational symmetry axis running through the longitudinal direction (Fig. 4a). In this scissor-shaped architecture the Bateman domains form the cutting blades and the ACT domains the handles with the center of β2 of the C-terminal CBS modules of both Bateman domains containing the pivot point. The CBS modules of the two Bateman domains are in a parallel arrangement in a head-to-head orientation meaning the N-terminal CBS1 domain of monomer one BsAcuB (chain A) interacts with the N-terminal CBS2 of the other monomer (chain B) and the same holds for the C-terminal CBS modules, i.e., CBS1' and CBS2' (Fig. 4a). The formation of an extended β-sheet was described for ACT domain containing proteins upon ACT domain dimerization[45]. In the BsAcuB dimer the two ACT domains of the two BsAcuB monomers are arranged in a side-by-side orientation, however, in this BsAcuB structure in complexes with two molecules of AMP and two molecules of ADP, the β-strands of the ACT domains are too far apart and not oriented appropriately to form a consecutive β-sheet (Fig. 4a). This might be different, however, when the ACT domains adopt a slightly different conformation. The structure of

BsAcuB reveals two adenine nucleotides per monomer in the Bateman domain, i.e., a total of four binding sites for adenine nucleotides (Fig. 4a). Next, we conducted a closer inspection of the nucleotide binding sites to understand the molecular mechanisms how adenine nucleotide binding might modulate AcuB's capacity to regulate AcuC activity.

## Electrostatics is important for regulation of AcuB function to modulate AcuC binding

Inspection of the crystal structure of BsAcuB bound to AMP and ADP shows the adenine nucleotides bind tightly to positively charged solvent-exposed cavities formed upon dimerization of the BsAcuB and accessible from both sides of the dimer (Fig. 4b). While the adenosine moiety is bound in a cavity open to the solvent the phosphate groups of AMP and ADP point towards the interior of the Bateman domains (Fig. 4b, c). The negative charges of the phosphates of the adenine nucleotides position, neutralize and/or shield several positively charged residues pointing from both monomers into the center of the cavities, i.e., Arg33, Arg51 and Lys120, otherwise experiencing electrostatic repulsion (Fig. 4b, c). Within each monomer the β-phosphates of the bound ADP molecules point towards the α-phosphates of the bound AMP molecules. The oxygens at the terminal phosphates are in a distance of 5.7 Å and 6.3 Å within chain A and chain B, respectively, and 7.1 Å and 6.6 Å across chain A and chain B in this ADP-AMP bound state suggesting the electrostatic repulsion is minimal due to their distance and the presence of the positively charged residues shielding the negative charges. However, the situation might be different if all four nucleotide binding sites were occupied by adenine nucleotides with more phosphate groups, i.e., ADP/ATP or ATP, resulting in steric and/or electrostatic repulsion effects with potential consequences on the conformation of the Bateman domains (Fig. 4b, c). The nucleotide binding sites on the corresponding faces in the BsAcuB dimer are equivalent, i.e., type I binding sites bind to ADP and type II binding sites to AMP in this BsAcuB structure (Fig. 4c). The molecular determinants mediating binding of the adenosine-5'-phosphate moiety is similar for all four binding sites (Fig. 4d). The ribose is bound by negatively charged side chains, i.e., Asp122 in type I binding sites and Asp52 in type II binding sites (Fig. 4c). The specificity for adenine nucleotides is created by specific interactions formed with the N⁶-exocyclic amino group of the adenine base with main-chain carbonyls of hydrophobic side chains, i.e., Ile12 for type I sites and Val83 for type II sites. Besides, Ile12/Val83 form further main chain interactions with their amide NH groups and N1 of the adenine purine ring (Fig. 4d). Additionally, the main-chain carbonyl of a second side chain, i.e., Arg33 in type I sites and Cys105 in type II sites, also bind the exocyclic amino group of the adenine base. All these interactions were not possible with a guanine base explaining the specificity towards adenine nucleotides. His34 constitutes an important residue acting as a wedge bridging to the α-phosphates of two nucleotide molecules bound by one Bateman

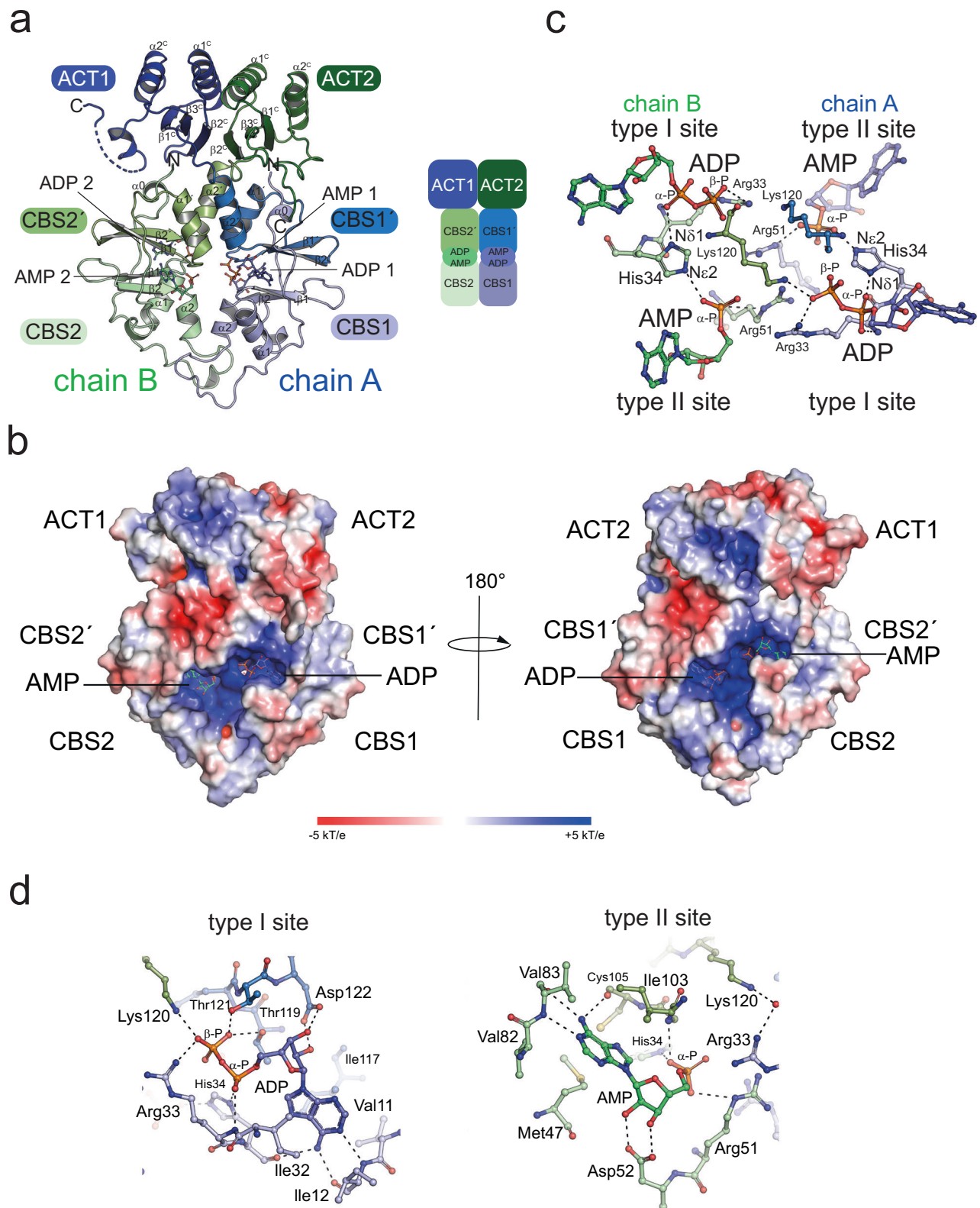

domain, i.e., ADP and AMP, with its imidazole $N_{\varepsilon 2}$ or $N_{\delta 1}$, respectively (Fig. 4c). Thereby, both adenine nucleotides bound in both binding sites in one BsAcuB monomer are directly connected independent of the type of adenine nucleotide. Next to His34 the α-phosphates of both, AMP and ADP, are bound by the $N_{\varepsilon}H$ of the guanidino group and the main chain amide of an Arg side chain, i.e., Arg33 in type I sites and Arg51 in type II sites. The adenine base is flanked by hydrophobic

residues, i.e., Ile32 and Ile117, in type I binding site and by Met47 and Ile103 in the type II binding site. Moreover, preference for ADP in the type I sites is created by several interactions binding the β-phosphate. This includes the side chains of Arg33, Thr119, Thr121, and Lys120. Notably, Lys120 is the only residue provided by the other monomer thereby stabilizing the interface and enabling a communication between both monomers in the BsAcuB dimer depending on the

**Fig. 4 | Crystal structure of BsAcuB in complex with two molecules AMP and two molecules ADP. a** Crystal structure of BsAcuB 2AMP 2ADP shows formation of a scissor-shaped dimer (PDB: 9SAV). A monomer consists of an N-terminal Bateman domain encompassing two CBS-modules and a C-terminal ACT domain (C: ACT domain). Each Bateman domain contains two adenine nucleotide binding sites, one of which is occupied by AMP and the other by ADP. The figure was created with PyMOL[88]. **b** The BsAcuB dimer contains two highly positively charged surface-exposed cavities binding to adenine nucleotides each. On each site of the dimer a binding site for AMP and ADP can be found. The adenosine moieties of the nucleotides are surface exposed while the phosphates point towards the interior of the Bateman domains. The electrostatics was calculated with the APBS Electro-statics Plugin in PyMOL[81,88]. The figure was created with PyMOL[88]. **c** Two adenine nucleotide binding sites are present in each monomer. Each monomer of AcuB

contains a type I binding site occupied by AMP and a type II binding site occupied by ADP for recognition of the adenine nucleotides. The adenosine moieties are bound by functionally similar interactions, the α-phosphates bound at both sides of an AcuB monomer are bridged by His34. Several positively charged residues, i.e., Lys120, Arg33 and Arg51, are neutralised and oriented by the adenine nucleotides. The figure was created with PyMOL[88]. **d** The type I and type II adenine nucleotide binding sites. The type I and type II adenine nucleotide binding sites provide a specific setup or residues for binding the adenine base, the ribose and the phos-phate moieties. Adenine nucleotides bound in one monomer are bridged by His34 binding the α-phosphates. The specificity towards adenine nucleotides is deter-mined by interactions with the exocyclic amino group at C6 and N1 of the adenine base (type I: main chain of Ile12, His34; type II: main chain of Cys105 and Val83). The figure was created with PyMOL[88].

---

nucleotide loading state (Fig. 4c). The physicochemical properties and arrangement of the residues in the nucleotide binding sites, the type of adenine nucleotide bound and their electrostatic and steric interplay might contribute to regulation of BsAcuB's capacity to modulate AcuC binding by affecting its conformation in solution. This is further explored in the next section.

## AcuB from *G. stearothermophilus* shows a similar overall struc-ture and nucleotide binding sites

To further understand how different adenine nucleotides affect AcuB function we switched to AcuB from the thermophilic *G. stearothermophilus*, i.e., GsAcuB, as we were not able to prepare nucleotide free BsAcuB due to its inherent instability. The genomic organization of the *acu*-operon encompassing *acuA*, *acuB*, and *acuC* and the location of *acsA*, encoding for AMP-forming acetyl-CoA synthetase, being reversely transcribed upstream of the *acu*-operon, as detected in *B. subtilis*, is also conserved in *G. stearothermophilus* suggesting similar mechanisms of regulation of AcsA activity by lysine acetylation and lysine deacetylation. We successfully crystal-lized GsAcuB, which was loaded to 15% with almost exclusively AMP (Fig. 1c). This enabled us to obtain crystals of GsAcuB in their nucleotide free state. The crystals belonged to the orthorhombic space group P2$_1$2$_1$2$_1$ and diffracted up to a resolution of 2.0 Å and contain one GsAcuB dimer per asymmetric unit (Fig. 5a; Supple-mentary Figs. 15, 16; Supplementary Table 4). The structure shows a high degree of overall similarity to BsAcuB bound to AMP and ADP showing each monomer consisting of an N-terminal Bateman domain consisting of two CBS modules, and a C-terminal ACT domain forming a scissor-shaped dimer (PDB: 9SAW; Fig. 5a).

We used these nucleotide free crystals for soaking experiments with various adenine nucleotides and combinations thereof, including AMP, ADP, AMP/ATP, ADP/ATP and Ap4A (Supplementary Table 4; Supplementary Figs. 17–28). This enabled us to solve crystal structures of GsAcuB in complexes with various adenine nucleotides (Supple-mentary Table 5). As described for BsAcuB, GsAcuB also shows four potential binding sites for adenine nucleotides in highly positively charged cavities within the Bateman domains (Fig. 5b). The adenine nucleotide binding sites are strongly conserved in GsAcuB compared to BsAcuB suggesting both proteins show a similar molecular mechanism underlying the regulation of AcuC activity (Fig. 5c).

All interactions described for nucleotide binding in BsAcuB can also be identified in GsAcuB, i.e., binding of the adenine base by hydrophobic interactions (type I site: Ile33, Ile118; type II site: Leu48, Ile104), creation of preference for adenine by interactions with the exocyclic N$^6$ amino group (type I site: main chain of Arg34, main chain of Ile13; type II site: main chain of Cys106, main chain of Ile84) and the N1 of the purine ring (type I site: main chain of Ile13; type II site: main chain of Ile84), binding of the 2'/3'-ribose hydroxyl groups by an Asp side chain (type I site: Asp53; type II site: Asp123) and binding of both adenosine α-phosphates within one monomer by His35 (Fig. 5c). In the molecules binding ADP nucleotides the β-phosphates in type I sites are

bound by interaction with two conserved Thr side chains, i.e., Thr120 and Thr121, and Arg34 as described analogously for BsAcuB. Notably, the structure of GsAcuB in complex with ADP shows Arg34 bridging both ADP molecules bound in type I and type II sites of each GsAcuB monomer by binding their β-phosphates (Fig. 5c). This shows His35 forms a wedge bridging the α-phosphates and Arg34 bridging the β-phosphates of two AMP or ADP molecules, respectively, bound to each Bateman domain of one monomer. A difference between GsAcuB and BsAcuB is Lys120 of monomer B in BsAcuB contacting the β-phosphate of the bound ADP in the type I site of monomer A. This Lys120 is replaced by a Gln121 in GsAcuB. Moreover, Arg51 in BsAcuB contacting the α-phosphates of the AMP molecules bound in the type II sites is replaced by Gln52 in GsAcuB (Fig. 4b, c; Fig. 5c). Overall, this shows the density of positive charge is lower in GsAcuB compared to BsAcuB which might be an adaptation contributing to thermostability.

## Conformational flexibility regulates AcuB function
The obtained structures of GsAcuB were all highly similar in their overall conformation independent of the nucleotide type bound (Fig. 5a; Supplementary Figs. 15–28). This might be due to the fact that we obtained the structures by soaking of nucleotide free GsAcuB crystals and crystal packing might hinder adopting grossly altered conformations (Fig. 5a; Supplementary Figs. 17–28). However, com-parison of the structures of GsAcuB in various nucleotide states with the structure of BsAcuB in complex with AMP and ADP enabled us to draw conclusions about the regulation of AcuB function dependent on the nucleotide loading state (Fig. 6).

We superimposed all individual AcuB Bateman domains, i.e., residues 1–132, including Bateman domains of BsAcuB monomers A to C and of all GsAcuB structures solved here, onto the Bateman domain of monomer A of nucleotide free GsAcuB (Fig. 6a, b). This shows an almost identical overall conformation of the Bateman domain dimer for all structures solved in this study (Fig. 6a, b; Supplementary Figs. 17–28). However, while all GsAcuB structures show a highly similar conformation of the ACT domains and a similar relative orientation of the ACT domains to the Bateman domains, the ACT domains of the three monomers of BsAcuB show a high degree of structural flexibility (Fig. 6b).

In all GsAcuB structures obtained by soaking of apo GsAcuB we obtained positive difference electron density at two regions directly in the interface of the two ACT domains lining helices α1 and α1' thereby bridging both ACT domains (Fig. 5a; Supplementary Fig. 15). A carboxylate group from formate or acetate could fit this density, but due to medium resolution and lacking further proof we con-servatively modeled these moieties as chloride ions alternating with two water molecules as the purification buffers contained high concentrations of potassium chloride (Fig. 5a). The negative charge is well stabilized by two main-chain interactions of Gly150 and Leu152. The tetrahedral coordination of the chloride is completed by side chain of Lys180 and a water molecule (Fig. 5a). The unas-signed anion might enable another mechanism of regulation by

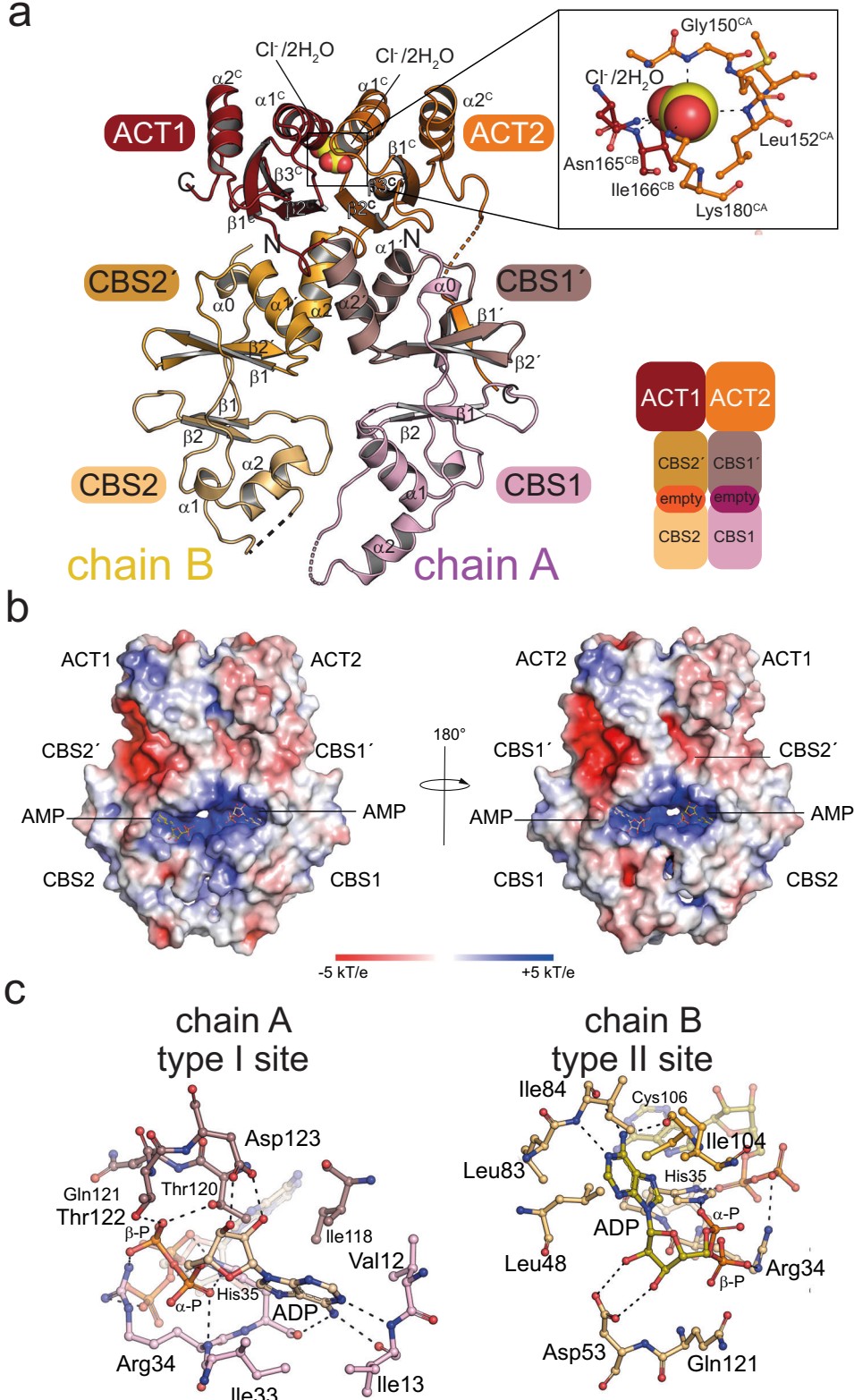

modulating the flexibility of the ACT dimer in AcuB. As we observed this density only in nucleotide free GsAcuB but not in BsAcuB bound to AMP and ADP the binding of nucleotides and binding of the unknown molecule in the ACT dimer might be mutually exclusive and lack of this molecule in BsAcuB might explain the high structural flexibility of the ACT domains. However, this needs further investigation.

Next to these differences in the ACT domains observed comparing nucleotide free GsAcuB and BsAcuB loaded with AMP and ADP, we made some observations showing the asymmetry of the GsAcuB monomers in the dimer. For the crystal structure of GsAcuB soaked with AMP all four nucleotide binding sites show full occupancy with AMP with B-factors in the range of 28–54 Å² (PDB: 9S52) (Supplementary Table 5). It is noteworthy that the B-factors for AMP bound to monomer A (type I site: 28 Å²; type II site: 35 Å²) are

**Fig. 5 | The GsAcuB contains adenine nucleotide binding sites as described for BsAcuB. a** Crystal structure of nucleotide free GsAcuB shows formation of a scissor-shaped dimer (PDB: 9SAW). A monomer consists of an N-terminal Bateman domain encompassing two CBS-modules and a C-terminal ACT domain (C: ACT domain). Each Bateman domain contains two adenine nucleotide binding sites, which are empty. In the ACT domain dimer two round-shaped positive electron densities lining the interface of α1-α1'. Two chloride ions (or alternatively four water molecules) were placed there. The figure was created with PyMOL[88]. **b** The GsAcuB dimer contains two positively-charged surface-exposed cavities binding to adenine nucleotides as shown here for the structure of GsAcuB in complex with four molecules of AMP (PDB: 9S52). As shown for BsAcuB the adenosine moieties of the nucleotides are surface exposed while the phosphate groups point towards the interior of the Bateman domains. The electrostatics was calculated with the APBS Electrostatics Plugin in PyMOL[81,88]. The figure was created with PyMOL[88]. **c** Two

types of adenine nucleotide binding sites can be distinguished in each monomer in GsAcuB as shown for BsAcuB. As representative the closeup shows the ADP binding sites for the structure of GsAcuB solved in complex with four molecules of ADP (PDB: 9SAX). The type I and type II adenine nucleotide binding sites provide a specific setup or residues for binding the adenine base, the ribose and the phosphate moieties. His35 forms a bridge by binding the α-phosphates of the adenine nucleotides binding on both sites of the AcuB monomer within the dimer. In the type I binding sites the of the dimer the β-phosphates of the bound ADP molecules interact with the side chains of Thr120 and Thr122 and of Arg34. Arg34 also binds the β-phosphate of the ADP molecule bound in the type II site. The specificity towards adenine nucleotides is determined by interactions with the exocyclic amino group at C6 and N1 of the adenine base (type I: main chain of Ile13, His35; type II: main chain of Cys106 and Ile84). The figure was created with PyMOL[88].

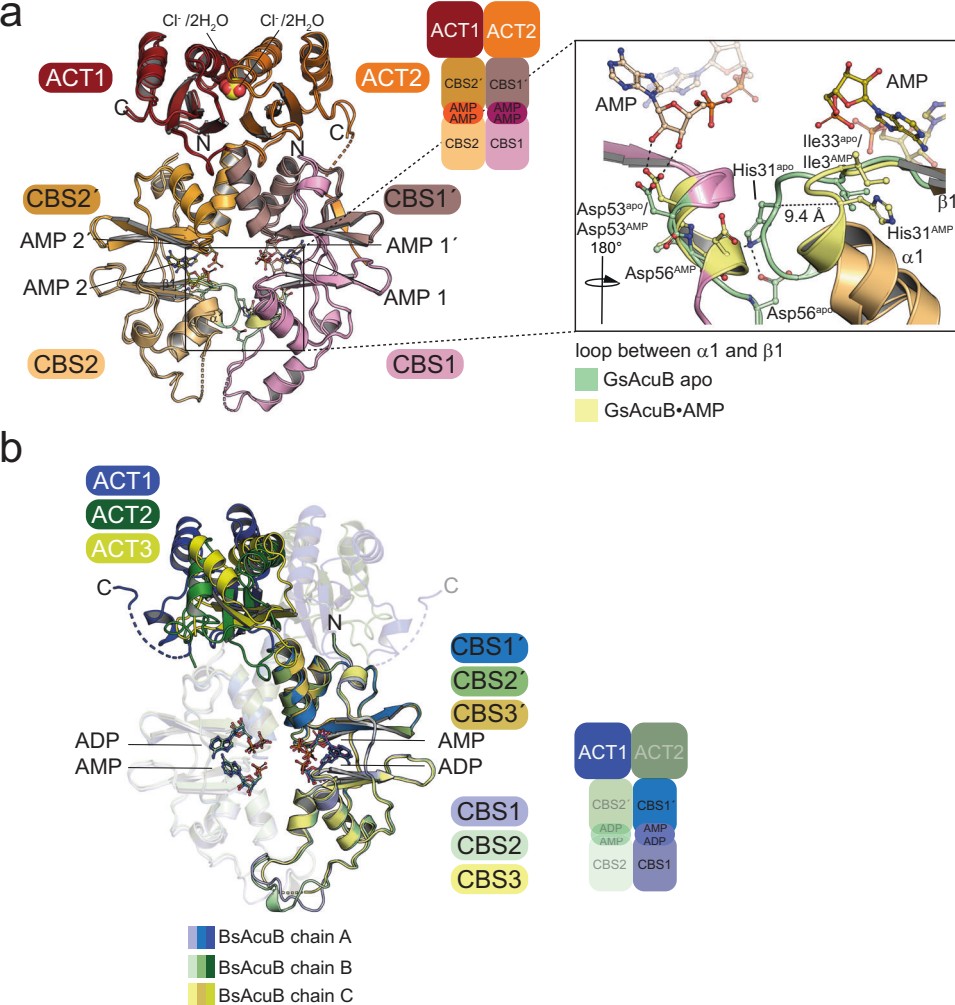

**Fig. 6 | GsAcuB forms an asymmetric dimer. a** To compare the structures of GsAcuB in its nucleotide free state and in complexes with various adenine nucleotides structural superpositions were conducted by SSM (secondary structure matching). Each monomer of all solved structures, i.e., monomer A or monomer B, was superimposed to the Bateman domain, i.e., aa 1–132, of monomer A of the structure of nucleotide free GsAcuB. Shown is the structure of AMP bound GsAcuB (PDB: 9S52) superimposed on nucleotide free GsAcuB (PDB: 9SAW) but all other structures behave similarly. Independent of the bound nucleotide type all structures are almost identical in the Bateman domains and the ACT domains. Formation of an asymmetric dimer was shown by comparison of nucleotide free GsAcuB and nucleotide bound GsAcuB. In the nucleotide free structure a substantial conformational change of the loop connecting α1 and β1 of the N-terminal CBS-domain of monomer B can be observed. This is exemplified by a 9.4 Å shift of the conformation of the Cα atom of His31.

This shows the AcuB dimer is asymmetric and binding of adenine nucleotides can result in conformational changes that might be important for AcuB function. The figure was created with PyMOL[88]. **b** The monomers of BsAcuB (PDB: 9SAV) show huge flexibility in the ACT domains. Each of the monomers of the structure of BsAcuB in complex with AMP and ADP, i.e., monomers A to C, were also superimposed on the Bateman domain, i.e., aa 1–132, of monomer A of the structure of nucleotide free GsAcuB (PDB: 9SAW). While the overall conformations of the Bateman domains of all monomers are structurally highly similar to the structure of nucleotide free GsAcuB, the ACT domains are highly flexible as are the orientations of the ACT domains to the Bateman domains. One difference is the presence of an unknown molecular entity in the ACT domains of nucleotide free GsAcuB, which might stabilize the ACT domains in GsAcuB. Whether this might explain the high flexibility in the ACT domains in AMP/ADP bound BsAcuB is not clear. The figure was created with PyMOL[88].

considerably lower compared to monomer B (type I site: 53 Å$^2$; type II site: 54 Å$^2$) with the type II site showing the highest B-factor. This indicates that the nucleotides bound to monomer A are better defined compared to monomer B. Moreover, this shows both monomers of the dimers are not identical and indicate in fact presence of asymmetry in the AcuB dimer. This trend can also be observed in the other GsAcuB structures soaked with different nucleotides and combinations thereof (Supplementary Table 5).

For all GsAcuB structures the nucleotide bound in the type II site in monomer B is the least well defined and shows the lowest occupancy values for the bound nucleotides. Except for GsAcuB soaked with AMP for all other structures it is even almost impossible to trace parts of the nucleotide apart from the adenine base at this site (Supplementary Fig. 5). For the structures of GsAcuB soaked with ATP and ADP we only obtained weak density for the γ-phosphate of ATP bound to the type II site of monomer A, while for crystals soaked with ATP and AMP we observed binding of three molecules of ADP while the type II site of monomer B does only show electron density for the adenine base (PDB: 9S4Z; 9S5O) (Supplementary Figs. 17–28). For all structures with a weak density for the nucleotide at the type II site of monomer B this suggests either only adenine/adenosine is bound due to hydrolysis of the ADP/ATP or the phosphates adopt different conformations with high degree of structural flexibility. As a support for the asymmetry of the AcuB dimer and the special features of the type II site in monomer B we made an interesting observation when comparing the structure of nucleotide free GsAcuB with the nucleotide bound structures of GsAcuB (Fig. 6a).

In the nucleotide free structure of GsAcuB the loop connecting α1 and β1 in monomer B makes a striking conformational change adopting an open state extending into the interior of the Bateman domain resulting in displacement of His31 by 9.4 Å compared to the nucleotide bound structures (Fig. 6a). This dramatic conformational movement of the loop in the nucleotide free state of GsAcuB might in part be driven by formation of a hydrogen bond between His31 and Asp56 as well as main chain carbonyl of Leu54. In the nucleotide bound conformations these interactions are not formed as nucleotide binding creates interactions with Asp53 and Ile33 both stabilizing the His31-containing loop in the closed state (Fig. 6a).

We also soaked nucleotide free crystals of GsAcuB with the stress molecule diadenosine-tetraphosphate (Ap4A). In contrast to all other nucleotide bound structures of GsAcuB/BsAcuB, His35/His34 does not act as wedge binding to the α-phosphates in the structure of GsAcuB in complex with Ap4A (PDB: 9SAY; Supplementary Figs. 20, 26). In fact, in the complex with Ap4A, His35 does not contact any of the Ap4A phosphates. Instead, in the type I site Arg34 contacts the Ap4A α-phosphate and Thr121 the β-phosphate while in type II site, the α-phosphate and β-phosphate are bound by Arg34 (Supplementary Fig. 21). Binding of the adenosine moieties is realised as described for the other structures. Our data shown above suggest AcuB is able to adopt different conformations dependent on the nucleotide loading state as represented by asymmetry in the AcuB dimer and the observation of local conformational differences in the AcuB dimer depending on the presence of adenine nucleotides. However, as our GsAcuB structures were obtained by soaking of nucleotide free crystals, we might not see the full repertoire of conformational dynamics within the AcuB dimer upon loading with different nucleotide types. The failure to co-crystallize GsAcuB with any of the nucleotides could be the consequence of a different conformation in solution. Deterioration of the apo crystals upon prolonged soaking goes along this line.

Overall, these data tempted us to speculate that the conformational dynamics within of the AcuB dimer is important to regulate binding to AcuC depending on the nucleotide loading state. To this end, we next performed analyzes by AlphaFold3 and mutational studies to characterize the complex formed between AcuB and AcuC.

## AcuB and AcuC form a heterotetrameric AcuB$_2$•2AcuC-complex

We were not able to obtain crystals for AcuB in complex with AcuC, neither for the complex of BsAcuB•BsAcuC nor for GsAcuB•GsAcuC, but our biochemical experiments show a direct interaction of AcuB and AcuC. Indeed, our data obtained by analytical SEC suggest AcuC and AcuB form a heterotetramer in the presence of AMP (Fig. 3b; Supplementary Fig. 9; Supplementary Fig. 10; Supplementary Table 1; Supplementary Table 3). To further validate the formation of a AcuB-AcuC-complex we performed enzyme assays to quantify the inhibitory potency of AcuB (Fig. 7a). These data show that GsAcuB is only capable to inhibit GsAcuC activity in presence of AMP, represented by the determined inhibitory constant, IC$_{50}$, of 38 nM (Fig. 7a). As in these experiments AcuB was used as an inhibitor for AcuC, the IC$_{50}$ value can be seen as a direct indication for the binding affinity of AcuB towards AcuC. Neither nucleotide free GsAcuB nor addition of ADP, ATP or Ap4A resulted in a GsAcuB able to potently inhibit GsAcuC to a comparable extent (Fig. 7).

To confirm these experimental data and to further investigate how AcuB binding to AcuC results in inhibition of AcuC activity and how binding of different adenine nucleotides modulates the capacity of AcuB to inhibit AcuC we generated an AcuB-AcuC complex by AlphaFold3. The prediction of the heterotetrameric BsAcuB$_2$•2ADP•2AMP•2BsAcuC complex is of high confidence with an ipTM+pTM-value of 0.85 + 0.85 forming a heart-shaped complex (Fig. 7b; Supplementary Table 7). The AlphaFold3 model supports that the C-terminal ACT domain is the major interaction site provided by BsAcuB to bind BsAcuC. The twofold-symmetry of the BsAcuB dimer causes AcuC contacting the ACT domain of one monomer but additionally is in interaction distance towards the Bateman domain of the other monomer (Fig. 7b). The contact region of AcuC towards AcuB is within the pivot region of the scissor-shaped structure suggesting a mode of regulation of BsAcuC binding to BsAcuB. This might be modulated by the relative position of the ACT domain towards the Bateman domain depending on different nucleotides bound to the Bateman domain. The AlphaFold3 model supports earlier reports in which the dimer of Bateman domains constitutes a sensor domain for adenine-containing molecules connected to an accessory domain, such as the ACT domain in AcuB, acting as effector domain[46–48].

We validated the AlphaFold3 model by performing mutational analyzes (Fig. 7c). We mutated M189 in the GsAcuA ACT domain (M188 in BsAcuB) to Arg, i.e., GsAcuB M189R, and H87 in the Bateman domain (H86 in BsAcuB) to Trp. i.e., GsAcuB H87W, to confirm that AcuC contacts both the ACT domain of one monomer and the Bateman domain of the other monomer of the AcuB dimer (Supplementary Fig. 29). These mutants were compared with GsAcuB wild type regarding their potency to form a complex with GsAcuC in presence of AMP and to inhibit its catalytic activity (Fig. 7c; Supplementary Fig. 30; Supplementary Table 8). These data revealed that GsAcuB M189R completely lost its capability to form a complex with GsAcuC, while GsAcuB H87W showed a reduced capability of complex formation compared to GsAcuB wild type (Supplementary Fig. 30; Supplementary Table 8). While GsAcuB showed an IC$_{50}$ value of app. 38 nM towards GsAcuC in presence of AMP, GsAcuB M189R was not capable to inhibit GsAcuC while GsAcuB H87R showed a moderately reduced IC$_{50}$ value of 71 nM (Fig. 7c). These mutational analyzes support the reliability of the AlphaFold3 model of the heterotetrameric AcuB$_2$•2AcuC-complex. Notably, binding of AcuB to AcuC blocks access of the substrate to the AcuC active site explaining mechanistically how AcuB inhibits AcuC activity (Supplementary Fig. 31).

We next compared AlphaFold3 models of BsAcuB and GsAcuB bound to various adenine nucleotides and combinations thereof

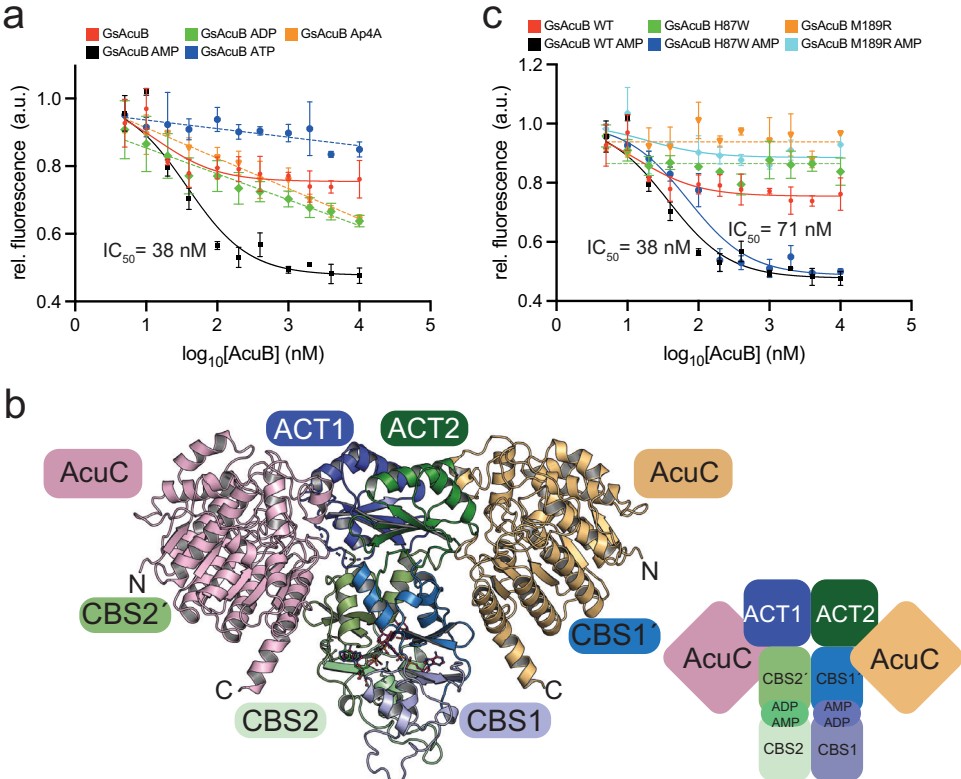

**Fig. 7 | The conformational dynamics in AcuB depends on the nucleotide-loading state and affects interaction of AcuC. a** GsAcuB inhibits GsAcuC in presence of AMP but not ADP, ATP, or Ap4A. IC50 values were determined by an FdL assay using Boc-Lys(Ac)-AMC as a substrate. The relative fluorescence of the released coumarin fluorophore was determined and is shown in arbitrary units (a.u.). The experiment was performed in three technical replicates (n = 3). Data are presented as mean ± SD. Source data are provided as Source Data file. **b** AlphaFold3 model of a BsAcuB$_2$•2BsAcuC heterotetramer with four molecules of AMP bound to the AcuB dimer. Two molecules of BsAcuC bind to the BsAcuB dimer. Each AcuC molecule forms major contacts to the ACT domain of one monomer and additionally contacts the Bateman domain of the other monomer. This creates an interaction site consisting of two contact sites of AcuC on both monomers of the

AcuB dimer, i.e., conformational changes in the scissor-shaped AcuB dimer either altering the distance of an ACT domain to a Bateman domain or their relative orientation might directly affect the binding affinity of AcuC towards AcuB. The figure was created with PyMOL[88]. **c** GsAcuB M189R is unable to inhibit GsAcuC and GsAcuB H87W has a reduced inhibitory potency compared to GsAcuB wild type. IC$_{50}$ values were determined by a FdL assay using Boc-Lys(Ac)-AMC as a substrate. The relative fluorescence of the released coumarin fluorophore was determined and is shown in arbitrary units (a.u.). The experiment was performed in three technical replicates (n = 3). The curves shown for GsAcuB wild type with AMP/without nucleotide are the same as shown in a. Shown are the mean-values and standard deviations. Source data are provided as Source Data file.

(Supplementary Figs. 32, 33). These data revealed similar overall conformations for the nucleotide bound proteins independent of the nucleotide loading state. However, we observed differences in the models of nucleotide free BsAcuB or GsAcuB compared to the nucleotide bound proteins (Supplementary Figs. 32, 33). AlphaFold3 is not capable to correctly predict the structures of ATP as well as Ap4A bound AcuB. Along that line, for the ATP bound AlphaFold3 model the γ-phosphates are too close (app. 1.9 Å; GsAcuB), and were even modeled within a covalent bond distance between the oxygens of the γ-phosphates of two ATP molecules present in type I and type II-binding site of one monomer (BsAcuB; Supplementary Figs. 32, 33). For the Ap4A bound models, AlphaFold3 predicts binding of each Ap4A molecules to each monomer while our experimental structural data shows binding to the nucleotide binding sites of two monomers within the AcuB dimer bridging two monomers. This is also supported by the nanoDSF-data showing the highest AcuB-stabilizing effect for Ap4A of the nucleotides compared (Fig. 1d; Supplementary Figs. 32, 33).

These results suggest that AlphaFold3 is not able to assess the conformational dynamics of AcuB binding to different nucleotides or combinations thereof. Our hypothesis is that binding of other adenine nucleotides than AMP with a higher number of phosphate groups, i.e. ADP and/or ATP, adds an electrostatic and steric component to the

system resulting in AcuB adopting different conformations. Only with AMP AcuB adopts a suitable conformation to bind AcuC. As these dynamics cannot be fully assessed by X-ray crystallography and AlphaFold3 model predictions, we next performed molecular dynamics (MD)-simulations to assess the conformational dynamics of AcuB dependent on the nucleotide-loading state.

## AcuB adopts different conformational states depending on the nucleotide-loading state

To investigate the stability and conformational landscapes of the AcuB dimer and AcuB•AcuC complexes, we performed a series of replica-exchange molecular dynamics (REMD) simulations using the TIGER2h$^{PE}$ approach. These simulations were initiated from Alpha-Fold3 models of a BsAcuB dimer and of the heterotetrameric BsAcuB$_2$•2BsAcuC complex with BsAcuB present in the nucleotide free state and charged with various nucleotides and combinations thereof, i.e., AMP, AMP/ADP and ATP (Fig. 7c; Supplementary Fig. 2; Supplementary Table 9). The molecular dynamics (MD) simulations should give information on the underlying mechanisms for AcuC-AcuB association and dissociation and to analyze whether this can be attributed to a certain AcuB nucleotide-loading states (Fig. 8a, b; Supplementary Table 10; Supplementary Fig. 34).

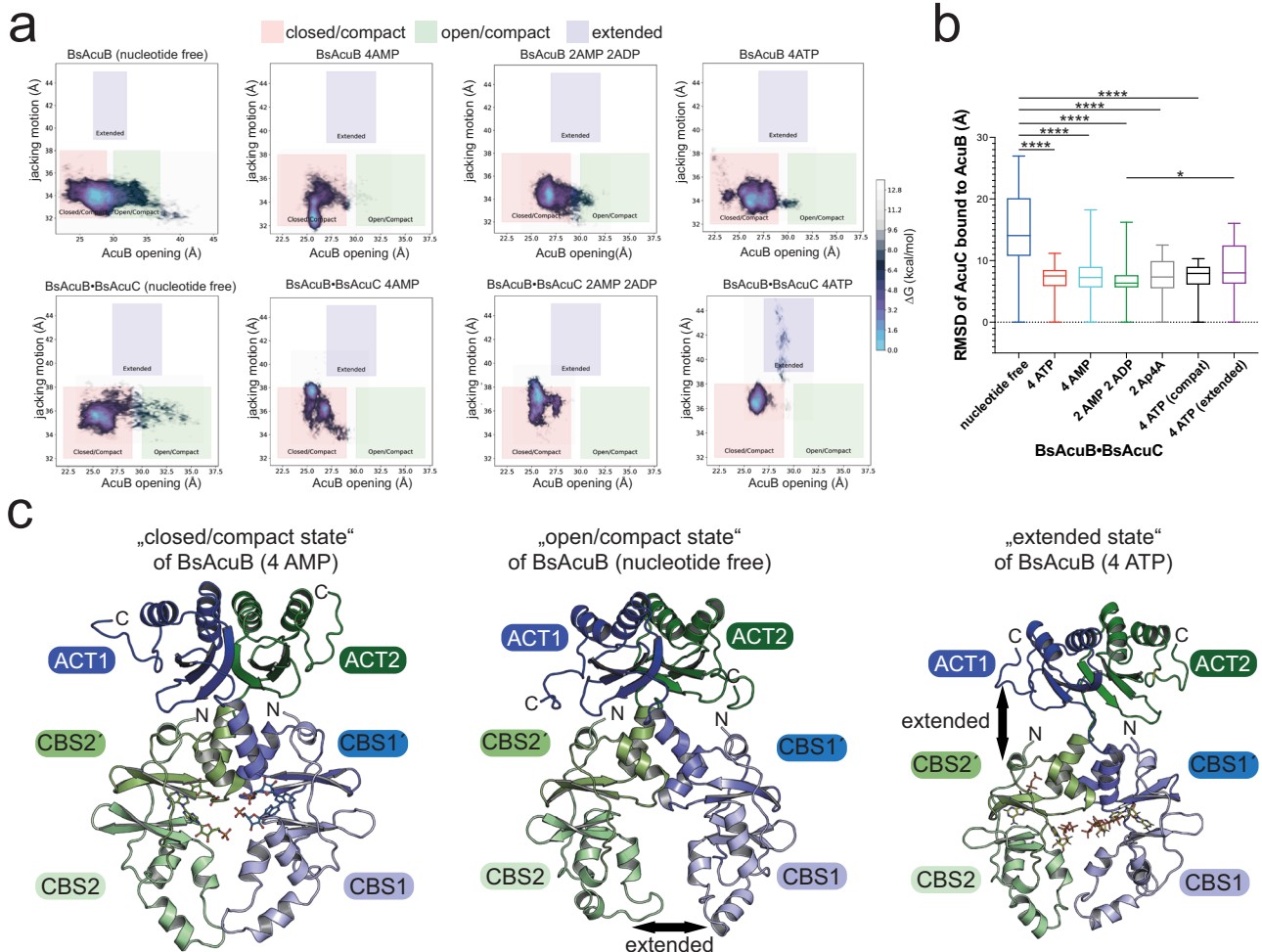

**Fig. 8 | Conformational dynamics in AcuB upon adenine nucleotide binding regulates its interplay with AcuC. a** Molecular dynamics simulations of the BsAcuB dimer and of AcuC bound AcuB. The calculations were initiated from an AlphaFold3 model of a dimer of BsAcuB or a BsAcuB₂•2BsAcuC heterotetramer. BsAcuB adopts three preferred conformational ensembles, i.e., "closed/compact" or "open/compact" ensembles with conformational dynamics within the BsAcuB Bateman domains and conformational ensembles in which the BsAcuB dimer performs a jacking motion along the longitudinal axes, i.e., "extended" ensembles. For AMP bound AcuB almost all conformations cluster in the "compact/closed" region, while AMP/ADP-bound BsAcuB adopts conformations more distributed to the "open/compact" region with higher conformational dynamics in the Bateman domain. The binding of ATP results in increase in "extended" conformations in complex with BsAcuC. In complex with BsAcuC binding of AMP to BsAcuB results in reduction of the flexibility of the Bateman domains mostly adopting "closed/compact" ensembles with an increase in the jacking motion adopting more "extended" conformational ensembles. **b** Comparison of dynamics of binding of BsAcuC to the BsAcuB dimer. Binding of nucleotide to BsAcuB results in lower RMSD values of BsAcuC binding to BsAcuB compared to nucleotide free BsAcuB. The uncorrelated RMSD values were determined after alignment of all structures in different nucleotide loading states to the BsAcuB dimer. The diagram shows a box-and-whisker plot depicting the median. The box represents the first and third quartile of the data and the whiskers min and max values. The statistical analyzes were performed conducting an ordinary ANOVA test ($p < 0.05$: *; $p < 0.01$: **; $p < 0.001$: ***; $p < 0.0001$: ****). Source data are provided as Source Data file. **c** Representative structures for the conformational clusters. Shown are structures of BsAcuB in complexes with BsAcuC. BsAcuC not shown for clarity. The structure of AMP bound BsAcuB represents a conformation of the "compact/closed" region with the Bateman domain in a closed conformation, the structure of nucleotide free BsAcuB represents the cluster of the "open/compact" region with huge conformational flexibility in the Bateman domain. The structure of ATP bound BsAcuB represents a conformation of the "extended" region. Shown are representative structures for each ensemble. The figure was created with PyMOL[88].

To systematically characterize these behaviors across all conditions, we computed center-of-mass distance distributions between specific domain groups using the following residue selections: (1) residues 1–82 of both BsAcuB monomers (N-terminal CBS domains), and (2) the ACT domain (residues 135–214) relative to the Bateman domain containing region (residues 1–134) of each AcuB monomer (Fig. 8a, b). These data revealed BsAcuB in the nucleotide free state being highly flexible particularly in the Bateman domains of BsAcuB (Fig. 8a, b). Furthermore, the interaction of BsAcuC to BsAcuB is highly dynamic in its nucleotide free state compared to nucleotide bound BsAcuB as represented by the calculated RMSD values of BsAcuC bound to BsAcuB (Fig. 8b).

To quantify these results for nucleotide free BsAcuB but also for complexes with various adenine nucleotides, we determined the occurrence of conformations of the BsAcuB dimer and the BsAcuB₂•2BsAcuC heterotetramer depending on the nucleotide-loading state of BsAcuB (Fig. 8a; Supplementary Fig. 34). We assessed the observed conformations by measuring the longitudinal distance distribution (jacking motion in Å) within BsAcuB in its uncomplexed state and in the BsAcuB₂•2BsAcuC complex and plotting these as a function of the transverse distance distribution indicating an opening of the two Bateman domains in the BsAcuB dimer (AcuB opening in Å). These studies revealed that BsAcuB clusters in different preferred conformational states dependent on the nucleotide loading (Fig. 8a; Supplementary Fig. 34). This enabled a

sorting of the observed conformations into three categories, i.e., the categories "closed/compact", "open/compact" and "extended" conformations (Fig. 8a; Supplementary Fig. 34). The nucleotide free BsAcuB dimer showed a large distribution of different conformations along the X-axis indicating the Bateman domains are very flexible and can dynamically adopt many open- and closed-states (Fig. 8a, c). However, the longitudinal distribution is well defined suggesting nucleotide free BsAcuB not adopting extended conformations (Fig. 8a).

These MD simulations support our experimental data showing that nucleotide free AcuB does not inhibit AcuC. This is also supported by root-mean-square deviation (RMSD) analysis on each of the complex simulations assessing the stability of AcuC binding to AcuB (Fig. 8b). This RMSD metric reflects the relative positional stability of AcuC with respect to AcuB. Higher deviations from the initial state indicate weaker binding or increased flexibility, allowing direct comparison of binding stability across different nucleotide-loading conditions (Fig. 8b). Apart from that, the MD simulations also revealed potential molecular mechanisms explaining the impact of AMP-loading of BsAcuB resulting in a protein capable to strongly bind and inhibit AcuC. For the simulations with nucleotide loaded BsAcuB we noticed that AMP and AMP/ADP stabilize the transverse conformational dynamics of BsAcuB (Fig. 8a, c). However, for AMP we observed that BsAcuB conformations accumulate in the "closed/compact" states more frequently compared to the AMP/ADP bound BsAcuB (Fig. 8a, c; Supplementary Fig. 34). For the BsAcuB$_2$•2AcuC complex the conformations observed in presence of AMP or AMP/ADP are similar adopting mostly "closed/compact" conformations in the Bateman domains, and showing slightly increased jacking motions compared to uncomplexed BsAcuB (Fig. 8a, c).

However, we made an interesting observation for the BsAcuB$_2$•2AcuC heterotetramer bound to ATP (Fig. 8a, c). As two ATP-molecules cannot bind simultaneously within one monomer of AcuB, the BsAcuB dimer adopts extended conformations when it is complexed with BsAcuC (Fig. 8a, c). The MD simulations show in this nucleotide loading state the Bateman domain preferentially exists in "closed conformations", while we observed accumulation of conformations showing a strongly increased longitudinal extension (Fig. 8a, c; Supplementary Fig. 34). In these conformations, the ACT domains are extending from the Bateman domains by stretching of the loops connecting the C-terminal CBS module with the ACT domain (Fig. 8a, c). This might result in a conformational state favoring dissociation of BsAcuC from BsAcuB, mechanistically explaining how ATP-loading of BsAcuB might result in displacement of BsAcuC from BsAcuB. This is one mechanism how BsAcuC selects the AMP bound conformation of BsAcuB for binding.

Our data suggest that there are minimal overall conformational differences in BsAcuB bound to AMP or AMP/ADP. However, our data support a model according to which AcuB can exist in many conformations in the nucleotide free state and nucleotide binding to AcuB stabilizes the conformations particularly within the Bateman domains.

As a summary, after examining the conformational ensembles of the different simulations, we identified two nucleotide-dependent structural rearrangements of interest. First, the lower domains of the BsAcuB Bateman domains of the dimer exhibited an opening motion, characterized by separation of the two monomers (Fig. 8a, b). This was most pronounced in the nucleotide free state. Second, in the BsAcuB$_2$•2BsAcuC complex with four molecules of ATP, the ACT domains BsAcuB detached from the core structure (Fig. 8a, b). Differences observed in binding of different nucleotides to AcuB and the consequences on its capacity to bind to BsAcuC might be due to stabilization of discrete conformations dependent on the type of adenine nucleotide. To fully understand the molecular mechanisms favoring BsAcuC binding to BsAcuB depending on the nucleotide loading further structural analyzes of the AcuB$_2$•2AcuC complex are needed.

## Discussion

Here we report structural and functional data on the regulation of AcuC deacetylase activity by the sensory protein AcuB. Our crystal structures of BsAcuB and GsAcuB show that they form a scissor-shaped dimer with each monomer structurally being composed of an N-terminal Bateman domain consisting of a tandem of CBS modules and a C-terminal ACT domain. In our structures the topology of the ACT domain is slightly distorted and the C-terminal β-strand is missing. This has been described also for other ACT domains in other proteins[45,49]. As reported for others the ACT domains from two AcuB monomers dimerize by interactions mediated by residues within α1. For some ACT domains the formation of an extended β-sheet was reported upon dimerization. This is not realised in AcuB - at least not in the structures in the adenine nucleotide loading states analyzed in this study. ACT domains were shown to bind to amino acids, metal ions and other small molecules and they are often found in proteins being part of the regulation of amino acid metabolism[45]. How binding of these small molecules is finally translated into a regulatory signal is still an unresolved question. Binding of small molecules or ions to ACT domains is suggested to stabilize a specific conformation and thereby reduce flexibility[45]. We observed that the ACT domains adopt a quite stable conformation in nucleotide free GsAcuB and in all nucleotide bound structures of GsAcuB solved here, while the monomers of BsAcuB in complex with AMP and ADP show a high conformational dynamics in the ACT domains and a relative orientation of the ACT domains to the Bateman domains.

In contrast to the crystal structure of BsAcuB in complex with two molecules of AMP and two molecules of ADP, we observed positive difference electron density in the interface of the two ACT domain monomers in the crystal structure of nucleotide free GsAcuB. As we were not sure which molecular entity is defined by this electron density we placed a chloride ion (or alternatively two water molecules) at these positions. However, a small organic acid such as acetate or formate would also well fit this electron density. This might indicate another regulatory mechanism by binding of a small molecule to the ACT domains thereby impairing their flexibility[45].

Bateman domains were found in several bacterial proteins, including AMP-activated protein kinase (AMPK) and inosine-5'-monophosphate dehydrogenase (IMPDH), acting as sensors for cellular adenine nucleotides or adenine containing molecules including coenzyme A or NAD$^+$[46,47,50,51]. In all of these enzymes these CBS modules containing sensor domains are connected to effector domains mainly mediating binding to effector proteins/enzymes, which are thereby regulated in their activity. Conformational changes in the Bateman domains upon binding of different nucleotides/adenine-containing molecules were suggested to affect binding of the effector proteins to the effector domain[47,48]. This in turn, allows adjusting the activity of the effector protein/enzyme to the cellular concentration of the sensed molecules.

We show here for BsAcuB and GsAcuB the Bateman domains are capable to bind a total of four adenine nucleotides, i.e., AMP, ADP or ATP, each monomer binding two nucleotides. Our crystal structures suggest that the Bateman domains are capable to bind simultaneously to an ADP and ATP molecule in one monomer, however, two molecules of ATP were not observed. This indicates that there is a limitation of phosphate groups due to electrostatic and steric effect that can bind to the type I site and type II site in one monomer of AcuB. The number of phosphate-groups in the nucleotides affects the conformation of the CBS module-containing Bateman domains exerting an electrostatic and steric effect and as a consequence binding to the deacetylase AcuC.

We also observe binding of AcuB to the *B. subtilis* stress molecule diadenosine-tetraphosphate (Ap4A). A crystal structure of GsAcuB in complex with Ap4A suggests binding of two Ap4A molecules per AcuB dimer with each Ap4A molecule occupying one type I site and one type

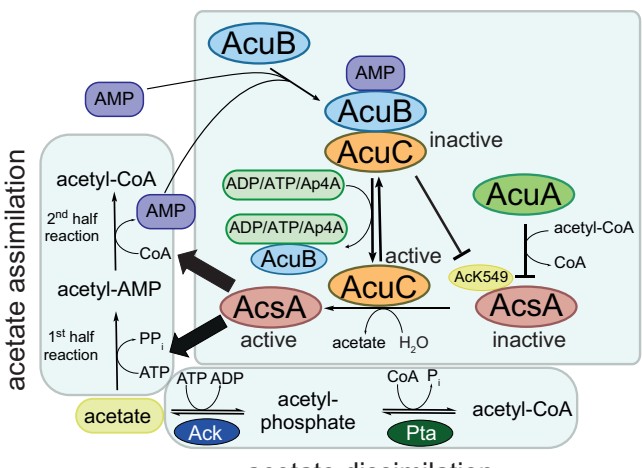

**Fig. 9 | Model for regulation of AcuC activity by the energy sensor AcuB with consequences on coordination of acetate assimilation and dissimilation.** AMP-forming acetyl-CoA synthetase AcsA is reported to mediate acetate assimilation to produce acetyl-CoA under conditions of low cellular glucose and moderate levels of acetate. Under conditions of high cellular acetate in the millimolar range acetate can be used in a second pathway of acetate assimilation to produce acetyl-CoA catalyzed by acetate kinase (Ack) and phosphotransacetylase (Pta). It is unknown how both pathways are coordinated and how AcsA activity is downregulated under conditions of high cellular acetate as observed under conditions such as overflow metabolism to ultimately ensure that not both of these metabolic pathways for acetate assimilation are simultaneously active. Here we report on the function of the protein AcuB as an energy sensor detecting the cellular energy charge inactivating AcuC activity under conditions of high cellular AMP thereby establishing an acetylated, inactive state of AcsA. In contrast, binding of AcuB to ATP or Ap4A does not inhibit AcuC. According to our model this allows to switch-off AcsA activity and to adjust its activity to the cellular metabolic state ensuring to be inactivated under conditions of high cellular acetate. Our study suggests AcuB is the missing link coordinating the activity of AcsA depending on the cellular energy charge and on the cellular acetate concentration as AMP is a by-product generated by AcsA upon production of acetyl-CoA from acetate. This constitutes another negative regulatory feedback mechanism allowing to switch-off AcsA activity under high concentrations of AMP initially accumulating by activity of AcsA under conditions of high cellular acetate. Thereby, AcuB plays a central role in coordination of acetate assimilation and dissimilation controlled by AcsA and the Ack/Pta-pathway under different cellular conditions. The figure was adapted from[29].

II site on the same face of the AcuB dimer thereby connecting both AcuB monomers. NanoDSF measurements showed that presence of all nucleotides, i.e., AMP, ADP and ATP, stabilize the AcuB protein, however, Ap4A has the strongest effect on AcuB stability. In *B. subtilis* the concentration of Ap4A was shown to increase under cellular stress conditions including oxidative stress or heat stress[42,52]. According to this, Ap4A has a dual function. Firstly, it stabilizes the AcuB protein under stress conditions including heat stress and, secondly, it abolishes the interaction of AcuB with AcuC thereby resulting in activation of AMP-forming acetyl-CoA synthetase activity ensuring production of acetyl-CoA under stress conditions. Aminoacyl-tRNA-synthetases, including the lysyl-tRNA synthetase, were reported to generate Ap4A[53]. Moreover, adenylate-forming enzymes including firefly-luciferase and AMP-forming acetyl-CoA synthetases were also shown to produce Ap4A from AMP-intermediates and ATP[54]. In this context, AcsA might be able to produce Ap4A and thereby re-activate AcuC under cellular stress conditions.

Our biochemical, structural biology data and MD simulations show that binding of AcuB to adenine nucleotides regulates its capacity to bind and to inactivate the deacetylase AcuC. This in turn allows to precisely adjust the activity of AcuC to the cellular energy charge. When AcuB is loaded with AMP, we observe binding and inhibition of

AcuC activity while nucleotide free, ADP- or ATP-loaded AcuB does neither bind to AcuC nor inhibit its activity. We provide a model according to which AcuB in its AMP-bound state is able to bind and to inactivate AcuC-deacetylase activity. As a consequence, under conditions of low cellular energy charge, i.e., high cellular AMP-concentrations, AcsA cannot be deacetylated and thereby not re-activated by AcuC. When cellular concentrations of ATP increase, loading of AcuB with ADP and/or ATP, disfavors binding to AcuC thereby being capable to deacetylate AcsA and re-activate it. This allows directly to adjust the activity of AcsA to the cellular energy charge, i.e., when the energy charge is low, acetyl-CoA synthesis catalyzed by AcsA is low but when the energy charge is high, acetyl-CoA is produced and can be used for biosynthesis. In that context it has to be stressed that the nucleotide free state of AcuB unlikely exists under physiological conditions as adenine nucleotides are always present. In fact, reports suggest that the ADP level in *B. subtilis* is relatively constant during growth while AMP is high in stationary phase and ATP is particularly high in exponential growth phase[55]. Therefore, changes in the AMP and ATP levels rather than changes in the ADP levels might regulate AcuB's capacity to interact with AcuC.

This nucleotide-dependent interplay between AcuB and AcuC gives a possible explanation for the observation that AcsA function is active and generates acetyl-CoA from acetate only under low acetate concentration. In contrast, it is inactive under very high millimolar acetate concentrations observed under conditions of carbon overflow resulting from acetate dissimilation, i.e., production of acetate from acetyl-CoA by Pta and Ack activity (Fig. 1b). Under these conditions the equilibrium of the reactions catalyzed by Ack and Pta can shift from acetate dissimilation towards acetate assimilation to produce acetyl-CoA thereby generating ADP. Moreover, initially also AcsA is active under these conditions of carbon overflow with high cellular acetate concentrations as the $K_M$-value of AcsA for acetate is in the micromolar range. However, the production of acetyl-CoA from acetate by AcsA produces AMP which results in loading of AMP to AcuB and inhibition of AcuC activity, which results in acetylated and inactivated AcsA (Fig. 9). This is another negative feedback mechanism that ensures that not both assimilative pathways are active, i.e., the AcsA pathway and the Ack/Pta pathway. It also ensures that the AcsA pathway is only active under low, micromolar concentrations of cellular acetate.

The conformational differences observed in AcuB upon binding different adenine nucleotide types are therefore likely of functional importance modulating AcuC binding to AcuB. Comparing the crystal structure of nucleotide free GsAcuB and AMP/ADP bound BsAcuB to other reported structures of Bateman domain containing proteins reveals the Bateman domains in AcuB are in a closed conformation in which the Bateman domains adopt a flat and regular disc-like structure possibly stabilized by association of molecules upon forming the crystal lattice. In this context it should be noted that the MD simulations demonstrated a quite high flexibility of nucleotide free BsAcuB in the Bateman domains. Our experimental approach soaking nucleotide free crystals of GsAcuB with selected nucleotides and nucleotide combinations might not show the full conformational changes occurring upon binding of different adenine nucleotides in solution. Any gross conformational changes in the GsAcuB dimer that occur upon nucleotide binding might result in damage to the crystal lattice. We were able to obtain diffracting crystals by soaking with those nucleotides. Therefore, we propose that the binding of those nucleotides did not result in major conformational changes in the GsAcuB dimer or alternatively the binding of the nucleotides is energetically not capable to interfere with the integrity of the crystal lattice. As a consequence, the nucleotides' β- and γ-phosphate moieties might adopt different conformations which would be observable in solution. If the conformations observed for nucleotide free GsAcuB and for nucleotide bound GsAcuB/BsAcuB in the crystal structures represent

the preferred conformations in solution needs further investigation. However, it is possible that binding of nucleotides to AcuB might induce different conformations or stabilize certain conformations by electrostatic or steric effects mediated by binding to different adenine nucleotides and this in turn modulates binding to the AcuC deacetylase.

Our data show that AMP-binding to AcuB is obligatory for AcuB to be capable to bind and inhibit AcuC. MD simulations support that AMP-binding to AcuB impairs the flexibility within the Bateman domains of each monomer by bridging both CBS domains. This is essential for the capacity of AcuB to inhibit AcuC. According to the conformational selection model for molecular recognition between proteins and ligands AcuB might exist in a multitude of pre-existing conformations with a high degree of structural flexibility[56–58]. Our data show that nucleotide binding to AcuB is a prerequisite to enable AcuC binding as nucleotide free AcuB does not bind AcuC. This suggests that inserting one phosphate moiety per nucleotide binding site in form of AMP is essential as counterions to neutralize the positive charges in the AcuB Bateman domains otherwise experiencing electrostatic repulsion. Whether all four nucleotide binding sites in AcuB have to be oocupied by AMP to be able to bind and to inhibit AcuC needs further investigation. Of all adenine nucleotides compared, the addition of AMP resulted in an AcuB dimer that was most potent to bind and to inhibit AcuC. Binding of AMP to the AcuB Bateman domains might select and stabilize conformation(s) of AcuB competent to be recognised and bound by AcuC. In such a model, the nucleotide would act as an allosteric modulator binding to the Bateman domains eliciting conformational changes propagating into the AcuC-binding region on AcuB[57,59,60].

As our data shows, AcuC binds mainly to the ACT domain but full inhibition is only observed with full length AcuB supporting the AlphaFold3 model according to which AcuC recognizes the ACT domain of one monomer and the Bateman domain of the other monomer. Binding of AMP to the Bateman domains of AcuB must stabilize a conformation of AcuB that is recognised by AcuC. If and how exactly binding of AMP to the sensory Bateman domains is transferred to the ACT effector domain needs further investigation. Overall, our data suggest that it is the decrease in flexibility of the Bateman domains and the relative orientation of the ACT domains to the Bateman domains occurring upon AMP binding that establishes the interaction site for AcuC binding. Mechanistically, the number of phosphate groups of the bound adenine nucleotides determines when this conformation is adopted as the bound adenine nucleotides do interfere sterically and electrostatically. Apparently, the steric and electrostatic effects of AMP in type I and type II sites in AcuB stabilize a conformation that can be bound by AcuC. The experimental data confirm that addition of ADP is sufficient to impair the interaction of AcuB to AcuC. This suggests binding of two adenine molecules per monomer with a total of more than three phosphate groups such as binding of two molecules of ADP, one ADP and ATP or two ATP molecules results in a conformation of AcuB incompetent to bind to AcuC due to electrostatic repulsion of the negative charges at the phosphates and/or steric effects.

As a summary, our data suggest that AcuB constitutes a sensor for the cellular energy charge (Fig. 9). Upon increase in cellular AMP concentrations AcuB is charged with AMP thereby stabilizing a conformation that can be bound by AcuC resulting in its inactivation. In contrast, binding of ADP and/or ATP to AcuB indicating a higher cellular energy charge disfavors binding of AcuB to AcuC. Moreover, also the stress molecule Ap4A is able to bind AcuB thereby stabilizing a conformation incapable to bind and inactivate AcuC. Binding of AcuB to AcuC results in inactivation of AcuC and ensures AcsA stays in an acetylated, inactive state ensuring acetyl-CoA production is low under conditions of low energy charge. Moreover, an indirect regulation of AcsA activity by the energy sensor AcuB allows a coordination of the AcsA-lysine acetyltransferase (AcuA) and -deacetylase (AcuC) activity and only a preferred AcsA activity under conditions of low glucose and low acetate concentration. This establishes another layer of regulation ensuring a tight control of AcsA activity and a coordination of cellular acetate assimilation and dissimilation by the AcsA and Pka/Ack-pathways. While well established in eukaryotes this is the first example for a bacterial classical deacetylase regulated by forming a protein complex.

## Methods

### Expression and purification of proteins

The *Bacillus subtilis* and *Geobacillus stearothermophilus* proteins BsAcsA (UniProt: P39062), BsAcuA (UniProt: P39065), BsAcuC (UniProt: P39067) and GsAcuC (EMBL: WP_237420713.1) were expressed as N-terminally His$_6$-tagged fusion proteins in pET-45b(+) in *Escherichia coli* BL21 (DE3) cells. GsAcuB (EMBL: WP_237420712.1) was expressed as C-terminally His$_6$-tagged fusion protein. The protein expressions were conducted in 2 L of Lysogenic Broth (LB) media. The synthetic genes were codon optimised for expression in *E. coli* (Biocat GmbH, Heidelberg). For expression, cells were cultivated to an OD$_{600}$ of 0.3 (37 °C; 130 rpm). Temperature was lowered to 18 °C and expression was induced by the addition of 0.2 mM of isopropyl-β-D-thiogalactopyranoside (IPTG). Cells were further cultivated overnight (18 °C; 110 rpm). Afterwards, the cells were harvested by centrifugation (4 °C, 6000 × *g*, 15 min) and resuspended in resuspension buffer (50 mM HEPES-KOH pH 7.5, 100 mM KCl, 2 mM β-mercaptoethanol with 0.2 mM Pefabloc protease inhibitor). The cells were lysed by sonication and the cleared lysate (20,000 × *g*, 60 min, 4 °C) was afterwards applied to Ni$^{2+}$-pre-charged HisTrap FF column equilibrated with the standard buffer (50 mM HEPES-KOH pH 7.5, 100 mM KCl, 2 mM β-mercaptoethanol). A washing step was performed with a high-salt washing buffer (50 mM HEPES-KOH pH 7.5, 500 mM KCl, 2 mM β-mercaptoethanol). The elution from the Ni-NTA column was done by applying a gradient of 0–1000 mM imidazole. For purification of GsAcuB (EMBL: WP_237420712.1), GsAcuB H87W and GsAcuB M189R a MgCl$_2$-containing standard buffer (50 mM TRIS/HCl, pH 7.4 (20 °C), 5 mM MgCl$_2$, 100 mM NaCl, 2 mM β-Me), washing buffer (50 mM TRIS/HCl, pH 7.4 (20 °C), 5 mM MgCl$_2$, 500 mM NaCl, 2 mM β-mercaptoethanol) and elution buffer (50 mM TRIS, pH 7.4 (20 °C), 5 mM MgCl$_2$, 100 mM NaCl, 1 M imidazole, 2 mM β-mercaptoethanol) has been used. For biochemical assays GsAcuB proteins were dialysed over night against standard HEPES buffer (50 mM HEPES-KOH pH 7.5, 100 mM KCl, 2 mM β-mercaptoethanol) using a 3 kDa-cutoff dialysis membrane. The *B. subtilis* AcuB (UniProt: P39066) was expressed from a pRSF-Duet-1 vector as C-terminally His$_6$-tagged fusion proteins in *E. coli* BL21 (DE3). The expression, lysis and IMAC was done as described above for the proteins from pET-45b (+). For the elution from the Ni$^{2+}$-NTA column an EDTA-buffer was used (50 mM HEPES-KOH pH 8.0, 500 mM KCl, 250 mM EDTA, 2 mM TCEP). A gradient was applied of 0–250 mM EDTA. Fractions of all proteins were collected and used for SDS-PAGE. Protein staining in SDS-PAGE gel was done with peqGOLD Marker (VWR Chemicals/Germany). The fractions of interest were pooled and concentrated with Amicon ultracentrifugation units (cutoff: 10 kDa, 4750 x *g*, 4 °C). Concentrated protein solutions of BsAcsA, BsAcuA, BsAcuC, GsAcuB and GsAcuC were further purified by size exclusion chromatography on a Superdex 200 16/600 or Superdex 75 16/600 column using the specific standard buffer, respectively. Fractions were collected and purity was analyzed by SDS-PAGE. The fractions containing the protein of interest were pooled and concentrated with Amicon ultracentrifugation units (cutoff: 10 kDa, 4750 × *g*, 4 °C). Concentrated proteins were shock-frozen in liquid nitrogen and stored at −80 °C until further use. Protein concentrations for AcsA, AcuA and AcuC were determined by measuring the absorption at 280 nm using the proteins' extinction coefficients. BsAcuB was

further purified by using PD-10 desalting columns (Cytiva). The standard gravity protocol by GE Healthcare (Instructions 52-1308-00 BB) was applied. For each fraction eluted from the column 5 μL were transferred onto a nitrocellulose-membrane and air dried. The membrane was stained with Ponceau S red solution to identify fractions with protein. Fractions with protein were pooled and concentrated with Amicon ultrafiltration units (cutoff: 10 kDa, 4750 g, 4 °C). Concentrated proteins were shock-frozen in liquid nitrogen and stored at −80 °C.

The *Bacillus anthracis* proteins BaAcuC (UniProt: A0A6H3AK00) and BaAcuB (UniProt: A0A6L8NZ01) were expressed and purified as their *Bacillus subtilis* equivalents by using a high salt standard buffer for purification steps (50 mM HEPES-KOH pH 7.5, 1000 mM KCl, 5 % glycerol, 2 mM ß-Me) with additional 10 mM imidazole for washing and 1 M imidazole (BaAcuC) or 250 mM EDTA (BaAcuB) for elution from Ni-NTA columns.

## Determination of the AcuB protein concentration and nucleotide loading state

Protein concentration for AcuB was determined firstly by RotiNanoquant (Roth) using commercially available bovine serum albumin (BSA) for calibration using the manual supplied by the manufacturer (Roth). Protein concentration for AcuB was also determined under denatured conditions. To this end, the absorption of AcuB protein samples was measured at wavelengths of 280 nm and 260 nm. Afterwards, the protein solution was denatured at 95 °C for 15 min. The samples were centrifuged at 13186 g for 15 min at 4 °C. The soluble phase was measured at 280 nm and 260 nm for reference. Protein concentrations were calculated by subtracting the 280 nm values by another Eq.(1). The difference was divided by the extinction coefficient of AcuB. Nucleotide concentration was determined by dividing absorptions at 260 nm after denaturing by the experimentally determined extinction coefficient for adenine nucleotides at 260 nm (AMP, ADP, ATP) Eq.(2). As a reference, the absorption at 260 nm was measured for pure adenosine nucleotide solutions (AMP, ADP, ATP) at a concentration of 1 mM.

$$c(AcuB) = \frac{A_{280nm}(before) - A_{280nm}(after)}{\varepsilon AcuB} \quad (1)$$

$$c(nucleotide) = \frac{A_{260nm}(after)}{\varepsilon AXP} \quad (2)$$

## Reversed phase-high pressure liquid chromatography (RP-HPLC)

For identifying nucleotides bound to AcuB and for determination of the nucleotide-loading of AcuB RP-HPLC experiments were performed. Here, a hydrophobic C18-reversed phase HPLC column was used as stationary phase. Equilibration was done using a phosphate-buffer (0.1 M Na$_2$H/PO$_4$/NaH$_2$PO$_4$, 10 mM TBAB (tetra-butyl-ammonium bromide), 7.5% (v/v) acetonitrile) as mobile phase. Protein samples were denatured at 95 °C for 15 min. The samples were centrifuged at 13,186 × g for 15 min at 4 °C. The soluble phase was diluted 1:10 by using filtered and degassed deionised water. 10 μL of each sample were injected onto a 4.6 × 150 mm extended C18-revesed phase column (Zarbax) with a flow rate of 1 mL/min at 30 °C. AMP, ADP, ATP standard curves were used for comparison if not stated otherwise.

## Analytical size exclusion chromatography (SEC)

To analyze the oligomeric states and the interaction of the AcuB and AcuC proteins, analytical SEC runs were performed on a Superdex 200 10/300 GL column (Cytiva). The SEC column was equilibrated with two column volumes (CV) of standard buffer (50 mM HEPES-KOH pH 7.5, 100 mM KCl, 2 mM TCEP). Before injecting the protein onto the column, the samples were incubated for 30 min on ice to ensure complex formation between AcuB and AcuC. Subsequently, 150 μl of 100 μM AcuC and 200 μM of AcuB with/without the addition of a 10 mM adenine nucleotides were injected. The runs were performed in standard HEPES buffer. If other concentrations were used or pre-incubation was done, it is indicated. Protein was detected by absorption at 280 nm and SDS-PAGE. Due to the low molar extinction coefficient (ε) of BsAcuB (MW: 25405 g/mol; ε: 8730 M$^{-1}$ cm$^{-1}$), GsAcuB wild type (MW: 24861 g/mol; ε: 9970 M$^{-1}$ cm$^{-1}$), GsAcuB H87W (MW: 24861 g/mol; ε:15470 M$^{-1}$ cm$^{-1}$) and GsAcuB M189R (MW: 24861 g/mol; ε: 9970 M$^{-1}$ cm$^{-1}$) compared to BsAcuC (MW: 44272 g/mol; ε: 78520 M$^{-1}$ cm$^{-1}$) and GsAcuC (MW: 43430 g/mol; ε: 80800 M$^{-1}$ cm$^{-1}$), complex formation was followed by the AcuC peak only. To monitor the elution of adenosine nucleotides, absorption at 254 nm was additionally recorded. To calculate the molecular weights of the proteins a calibration curve was determined using standard proteins (ribonuclease A (13.7 kDa,), carbonic anhydrase (29 kDa), ovalbumin (44 kDa), covalbumin (75 kDa), aldolase (158 kDa) and thyroglobulin (669 kDa)) from the low and high molecular weight calibration kit (Cytiva). A run with blue dextran was performed to calculate the void volume of the column. The partition coefficients $K_{av} = (V_e - V_0)/(V_c - V_0)$ ($V_e$ elution volume, $V_0$ void volume, $V_c$ column volume) were plotted against the log molecular weight. The resulting calibration equations are shown together with the coefficient of determination ($R^2$) showing the accuracy of the fit.

## Structure prediction using AlphaFold3

To analyze the effect of different adenine nucleotides on the conformation of AcuB alone or on complex formation between AcuB and AcuC AlphaFold3 version 3.0.0 was used (https://github.com/google-deepmind/alphafold3/releases/tag/v3.0.0) was used. Sequences of BsAcuB (UniProt: P39066) and BsAcuC (UniProt: P39067) were used for the predictions. Models were scored using pLDDT (predicted local distance difference test) for monomers and ipTM (interface ptm) + pTM (predicted TM) scores for the complexes. The model with the highest overall pLDDT or ipTM + pTM scores were used for further analyzes.

## Molecular dynamics (MD) simulations

**TIGER2H$^{PE}$ simulations.** To investigate the stability and conformational landscapes of the AcuB dimer and AcuB•AcuC complexes, we performed a series of replica-exchange molecular dynamics (REMD) simulations using the TIGER2h$^{PE}$ approach (Supplementary Table 10)[61]. This method is particularly well-suited for large biomolecular systems that would otherwise be computationally prohibitive for conventional REMD, as it decouples exchange phases from the systems degrees of freedom. This enables efficient exploration of conformational space and collection of representative structural ensembles, even for large protein complexes, that are unattainable by conventional MD simulations. Using TIGER2h$^{PE}$, we systematically compared different nucleotide-loading states to identify their effects on the dynamic behavior and distribution of conformational substates within the complexes.

Simulations were conducted with NAMD 2.14[62], using standard simulation settings[63], including the TIP3P water model and hydrogen mass repartitioning (HMR) on the solute[64], which allowed for an increased integration timestep of 4 fs. Systems were constructed via CHARMM-GUI[65] and parameterised with the CHARMM36 force field. Parameters for ATP, ADP, AMP, and AP4A were obtained from the CHARMM General Force Field (CGenFF) via the CHARMM Small Molecule Library[66], and used net charges of −4, −3, −2, and −4, respectively. System neutrality was ensured by adding appropriate numbers of Na$^+$ or Cl$^-$ ions.

Each system was solvated in a cubic box with side lengths of 80 Å for AcuB only simulations and 120 Å AcuB•AcuC complexes. To

focus sampling on local conformational substates, flat-bottom RMSD restraints with a threshold of 8 Å and a force constant of 50 kcal/mol/Å² were applied to the backbone atoms of each AcuB and AcuC molecule, separately, with prior alignment of the reference structure. This helped to limit global rearrangements within one protein, while still allowing for internal flexibility, without preventing dissociation or refinement of the complex.

TIGER2h$^{PE}$ simulations were run with eight replicas, spanning a temperature range from 310 K to 450 K. Each REMD cycle consisted of 16 ps of sampling followed by 8 ps of cooling. Replica exchanges were attempted at the end of each cycle. The temperature ladder was optimised to ensure appropriate exchange rates and temperature diffusion.

Convergence of conformational sampling was evaluated by monitoring backbone RMSD of the AcuB dimers and confirming a stable 500-frame running average without directional drift, indicating stationary and recurrent sampling consistent with locally ergodic behavior (Supplementary Fig. 35).

**AcuB conformational plasticity.** After examining the conformational ensembles of the different simulations, we identified two nucleotide-dependent structural rearrangements of interest. First, the lower domains of the AcuB dimer exhibited an opening motion, characterized by separation of the two monomers. Second, in the BC4T condition, the head domain of AcuB detached from the core structure. To systematically characterize these behaviors across all conditions, we computed center-of-mass distance distributions between specific domain groups using the following residue selections: (1) residues 1–82 of both AcuB monomers (N-terminal CBS module), and (2) the ACT domain (residues 135–214) relative to the Bateman domain core region (residues 1–134) of each AcuB monomer.

These distance metrics were computed using Python and the MDAnalysis library[67]. Joint probability distributions over these two degrees of freedom were calculated and converted into free energy surfaces using the relation:

$$\Delta G = -RT \cdot \ln\left(\frac{P}{\max(P)}\right) \qquad (3)$$

where R is the gas constant, T is the temperature, and P is the normalized probability density. The resulting free energy landscapes were analyzed using the density-based OPTICS clustering algorithm (from scikit-learn) to identify distinct conformational states.

This analysis revealed three major conformational basins, which we termed:

- compact/closed—corresponding to the initial state
- compact/open—representing partial opening of the AcuB dimer
- extended—a distinct configuration in which the AcuB headpiece is displaced, possibly related to a functional ejection mechanism for AcuC.

**AcuC binding stability via RMSD.** To assess the stability of AcuC binding to AcuB, we performed root-mean-square deviation (RMSD) analysis on each of the complex simulations. All analyzes were carried out using the MDAnalysis library in Python[67]. For each condition, the full conformational ensemble was loaded, and trajectory frames were aligned to the first frame using non-hydrogen atoms from both AcuB chains. RMSD was then calculated for the complementary segments of both AcuC chains, using the same reference frame.

This RMSD metric reflects the relative positional stability of AcuC with respect to AcuB. Higher deviations from the initial state indicate weaker binding or increased flexibility, allowing direct comparison of binding stability across different nucleotide-loading conditions.

While replicas in TIGER2h$^{PE}$ already represent independent timelines, to mitigate statistical bias due to temporal correlation within the merged state ensembles, we computed the autocorrelation function (ACF) of each RMSD time series. From the ACF, we derived the integrated autocorrelation time (IAT), which estimates the minimum number of frames required between samples to ensure statistical independence. RMSD values were then sub-sampled using the IAT to obtain uncorrelated datasets for each condition.

For each simulation, we calculated the mean and standard deviation of the uncorrelated RMSD values. Statistical comparisons between nucleotide-loading states were conducted using ordinary one-way ANOVA test with Tukey test to correct for multiple comparisons using statistical hypothesis testing and significance was evaluated using standard thresholds ($p < 0.05$: *, $p < 0.01$: **, $p < 0.001$:***; $p < 0.0001$:****). Results were visualised with boxplots annotated with corresponding significance levels.

### Crystallization, data collection and refinement

Crystals for AcuB of *B. subtilis* (UniProt: P39066) were obtained by the sitting drop vapor diffusion method in the crystallization condition containing 5% (v/v) MPD, 5% (v/v) ethanol, 0.1 M HEPES-KOH pH 7.6 using 13 mg/mL of protein. The crystals belonged to the orthorhombic space group P2$_1$22$_1$ with three molecules per asymmetric unit. Crystals were grown at 20 °C in 24 h. For cryoprotection of the crystals before cryocooling in liquid nitrogen additional 15% (v/v) MPD were added. The native data set was collected at the HZB/BESSY II Berlin/Germany at 100 K on beamline 14.1 at a wavelength of 0.976 Å using a PILATUS3 S 6 M detector. The asymmetric unit contains three monomers with an AMP and an ADP each resulting in 2 independent dimers. Crystals for nucleotide free AcuB of *G. stearothermophilus*, GsAcuB, were obtained by the sitting drop vapor diffusion method in the crystallization condition containing 10% (w/v) PEG20000, 0.1 M MES pH 6.5. For cryoprotection of the GsAcuB crystals additional 15% (v/v) PEG400 were added before freezing in liquid nitrogen. The crystals belonged to the orthorhombic space group P2$_1$2$_1$2$_1$ with two molecules per asymmetric unit. Soaking was performed by adding 10 mM of nucleotides to the crystallization condition. Crystals were incubated overnight. For cryoprotection 15% (v/v) PEG400 were added. The native data set was collected at the DESY Hamburg/Germany at 100 K on beamline P13 at a wavelength of 0.918 Å using an EIGER 16 M detector. The oscillation range was 0.1° and 3600 frames were collected. Processing, i.e., indexing and integration, of the datasets was done using the XDS package[68,69]. For subsequent steps the CCP4 program suite (v7.1.0.18 or v8.0.0.19) was used[70–72]. Scaling and merging were performed with AIMLESS (version 0.3.8)[73,74]. Initial phases were determined using the program Phaser within the CCP4 program suite[75]. The AlphaFold3 structural model of BsAcuB or GsAcuB was used as the search model, respectively[76,77]. Refinement was done using the program REFMAC5 using NCS and TLS[78]. COOT (version 0.9.6) was used for model building into the 2F$_o$ − F$_c$ and F$_o$ − F$_c$ electron density maps in iterative rounds of refinement with REFMAC5[78,79]. Validation of the structural models by MolProbity shows all residues are in or close to the preferred and allowed regions of the Ramachandran plot[80]. All structure figures were made with CCP4mg 2.11.0, PyMOL 3.0, and PyMOL 2.5.4[81]. Data collection and refinement statistics are given in Supplementary Table 1. R$_{work}$ is calculated as follows: R$_{work}$ = $\sum |F_o − F_c|/\sum F_o$. F$_o$ and F$_c$ are the observed and calculated structure factor amplitudes, respectively. $R_{free}$ is calculated as $R_{work}$ using the test set reflections.

### Fluor-de-Lys (FdL) assay to determine the AcuC deacetylase activity

For screening of deacetylase activity toward the Boc-Lys(Ac)-AMC (Sigma-Aldrich, cat no. SCP0168) the Flour-de-Lys assay was used. The

relative comparison was performed by using 20 nM of AcuC, 20–400 nM of AcuB, 100 μM Boc-Lys(Ac)-AMC (sigma) and 1 mM of adenosine-nucleotides. All samples were incubated at 25 °C for 60 min in standard buffer (50 mM HEPES-KOH pH 7.5, 100 mM KCl) using 2 mM of reducing agent (β-mercaptoethanol, DTT). Samples were incubated in black med.-binding 96-well microtiter plates (F-bottom, chimney well, Greiner Bio-One, cat no. 655076) and the reaction stopped by adding stop-solution (10 μM TSA (Sigma-Aldrich, cat no T8552), 2 mg/mL trypsin (Sigma-Aldrich, cat no T4799)), and incubated for 30 min at 37 °C. Samples were measured at wavelength of 360 nm for excitation and recorded at 440 nm for emission (Infinite® 200 PRO, TECAN).

### Fluor-de-Lys (FdL) assay to determine IC50 values for GsAcuB wild type and mutants

Flour-De-Lys assays were performed using GsAcuC and GsAcuB wild type and mutants. Assays were conducted in standard HEPES buffer (50 mM HEPES-KOH pH 7.5, 100 mM KCl, 2 mM β-mercaptoethanol) using 200 μM Boc-Lys(Ac)-AMC (sigma) and 1 mM of adenine nucleotides. For the determination of $IC_{50}$ 20 nM of GsAcuC was titrated with increasing concentrations of GsAcuB or mutants thereof (5 nM to 10 μM) without addition of nucleotides and with addition of adenine nucleotides (AMP, ADP, ATP, Ap4A). GsAcuB proteins were incubated at room temperature for 15 min with nucleotides. The reaction was started by adding GsAcuC. All samples were incubated at room temperature for 30 min in black 96-well microtiter plates (F-bottom, chimney well, Greiner Bio-One, cat no. 655076) and the reaction was stopped by adding stop-solution (100 μM SAHA (Sigma-Aldrich), 2 mg/mL trypsin (Sigma-Aldrich, cat no T4799)) and further incubated for 45 min at room temperature. Samples were measured at wavelength of 340 nm used for excitation and fluorescence emission was recorded at 460 nm (Infinite® 200 PRO, TECAN). For normalization the maximum fluorescence signal was set as maximum value and lowest detected fluorescence signal was used as minimum. Normalized values for each data point were calculated using Eq.( 4). For calculation of $IC_{50}$ values a sigmoidal dose-response inhibition model was applied using GraphPad (dose-response model with Hill slope set to −1) Eq.(5). If sigmoidal fit was not suitable, we used linear regression to indicate the trend.

$$\text{rel.fluorescence} = \text{minimum} + \frac{\text{fluorescence} - \text{minimum}}{\text{maximum} - \text{minimum}} \qquad (4)$$

$$\text{fit.fluorescence} = \text{minimum} + \frac{\text{maximum} - \text{minimum}}{1 + 10^{\log(IC50 - X)*\text{HillSlope}}} \qquad (5)$$

### Immunoblotting

For immunoblotting, AcuB (50 μM) proteins were incubated with/ without 1 mM of nucleotides for 15 min on ice. Afterwards, AcuC (2.5 μM) were added and incubated for another 15 min on ice. Lastly, acetylated BsAcsA (15 μM) was added and samples were incubated for 15 min at RT. Reactions were stopped with Laemmli Buffer (50 mM Tris/HCl pH 6.8, 50% (v/v) glycerol, 10% (w/v) SDS, 0.5% (w/v) bromophenol blue, 7% (v/v) β-mercaptoethanol) and denaturing at 95 °C for 10 min. The samples were separated by SDS-PAGE and afterwards, the proteins were transferred to a nitrocellulose-membrane using a semi-dry immunoblotting system (Biorad; 48 min, 12 V, 200 mA). Afterwards, the membrane was stained with Ponceau S-red solution to analyze the completeness of the transfer and as a loading control. The membrane was blocked with 5% (w/v) semi-skimmed milk PBS-T (1 h, room temperature) and afterwards incubated with the primary anti-acetyl-lysine antibody (anti-AcK-AB; abcam ab21623, 1:2000 in 5% w/v semi-skimmed milk PBS-T; overnight, 4 °C). The membrane was washed three times with PBS-T buffer (10 min, room temperature) before the addition of the HRP-coupled secondary antibody (goat anti-

rabbit-AB; abcam ab6721 1:5000 in 5% (w/v) semi-skimmed milk PBS-T). Detection was done by using enhanced chemiluminescence (Roth) with luminol. All uncropped images are provided in the Source Data file.

### Nano-differential scanning fluorimetry (nanoDSF)

For analysing melting temperatures, $T_m$, of proteins and the protein stability dependent on presence of different adenine nucleotides nanoDSF experiments were performed using *B. subtilis* and *G. stearothermophilus* AcuB proteins (Prometheus NT.48 NanoDSF; NanoTemper Technologies GmbH). For identification of melting temperatures purified proteins were analyzed at concentrations of 1 mg/mL (ca. 40 μM AcuB; ca. 23 μM AcuC) together with 1 mM of selected adenine nucleotides (AMP, ADP, ATP, AP4A). For analyzes of interactions 0.5 mg/mL of GsAcuC and 2 mg/mL GsAcuB were used. Samples were incubated for 30 min on ice before each measurement. High sensitivity capillaries (NanoTemper Technologies GmbH) were filled with 10 μL of samples. A temperature gradient was applied from 15 °C up to 95 °C with temperatures rising with a rate of 1 °C/min. Fluorescence intensity (I) was measured at 330 nm and 350 nm with 50% excitation power for samples with AcuC and 100% excitation power for samples with only AcuB with/without addition of adenine nucleotides. For interaction analysis excitation power was set between 25–30%. $I_{350\,nm}/I_{330\,nm}$ was plotted against the temperature. Melting temperatures were defined as extrema of the first derivative of the ratio as calculated by the instrument software (Prometheus NT.48 NanoDSF; NanoTemper Technologies GmbH) (Eq. 6).

$$T_m = \max \text{ or } \min(d(I_{350nm}/I_{330nm})/dT) \qquad (6)$$

Interaction between GsAcuB and GsAcuC was tested with $I_{330\,nm}$ data. For non-interacting molecules intensities and their derivatives dI/dT are additive. The difference between the measured derivatives for GsAcuB-GsAcuC mixtures and the best linear fit by a linear combination of the individual derivatives was taken as a measure for interaction.

### Sequence- and structure alignments

All protein sequences used for alignments were obtained from UniProt database with the accession codes shown above[82,83]. Sequence alignments were conducted with the multiple sequence alignment (MSA) program Clustal Omega (v0(1.2.4)). Further analysis was done using the software ESPript3.0 (v3.0.10)[84,85]. As structure models either obtained crystal structures obtained for AcuB or AlphaFold3 models for AcuC were used[77]. For structural alignments, crystal structures obtained for AcuB in this study and AlphaFold3 models obtained for complexes of AcuB and AcuC were used. The alignment was performed using PyMOL[81].

### Data analysis and visualization

For quantitative analyzes of the results obtained for the immunoblots Fiji (ImageJ 2.0.0-rc-68/1.52 h) was used[86]. The raw data from most experiments was initially processed using Microsoft Excel 2011. Further data visualization and statistical analyzes was done by GraphPad Prism version 8 or version 9.5.1. SnapGene Viewer 5.1.4.1 was used for handling of DNA sequences and generation of plasmid maps (SnapGene software from Insightful Science; available at snapgene.com). PyMOL version 2.5.3 and PyMOL version 2.3.4 were used to visualize protein structures, surface electrostatics and electron density maps[87]. ChemDraw version 23.0.1 was used to draw chemical structures of nucleotides. Adobe Photoshop 22.3.1 and Adobe Illustrator 25.4.1 were used to create figures.

## Statistics and reproducibility

(De-)acetylation assays were performed in at least two independent experiments as indicated with similar results obtained. Analytic size-exclusion chromatography and subsequent SDS-PAGE of AcuB, AcuC, and complexes thereof was done in duplicate or triplicate with showing one example of respective elution fractions. For X-ray crystallographic structure determination of AcuB in various nucleotide-loading states, data sets were collected from single crystals without merging of data sets. All assays were performed in independent replicates. For bar graphs, the standard deviations (SD) and mean values were shown. No statistical method was used to predetermine sample size. Unpaired, two-tailed student's t-tests were performed to assess statistical significance with significance levels as indicated.

## Reporting summary

Further information on research design is available in the Nature Portfolio Reporting Summary linked to this article.

## Data availability

The coordinates and structure factors for the structures of AMP-ADP loaded BsAcuB, (PDB: 9SAV), nucleotide free GsAcuB (PDB: 9SAW), AMP-loaded GsAcuB (PDB: 9S52), ADP-ATP loaded GsAcuB (PDB: 9S51, [https://doi.org/10.2210/pdb6GZS/pdb]), AMP-ATP-loaded GsAcuB (PDB: 9S4Z), AMP-ATP-loaded GsAcuB (PDB: 9S50), ADP-loaded GsAcuB (PDB: 9SAX), Ap4A-loaded GsAcuB (PDB: 9S4Y, [https://doi.org/10.2210/pdb9SAY/pdb]) were deposited in the PDB (http://www.rcsb.org). The source data underlying Figs. 1, 2, 3, 7, 8 and Supplementary Figs. 1, 3, 6, 8, 9, 10, 11, 30 are provided as a Source Data file. Uncropped images of Supplementary Figs. are shown at the end of the Supplementary Information. The initial configuration, full conformational ensembles (trajectories) from TIGER2h$_{PE}$ simulations and major clusters are provided at Zenodo [https://doi.org/10.5281/zenodo.16780123]. Source data for this study are provided with this article and its Supplementary Information. Source data are provided with this paper.

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

## Acknowledgements

This work was supported by the German Research Foundation (Deutsche Forschungsgemeinschaft DFG) with a research grant proposal with the grant no. 534243417 (LA2984/8-1) and the major instrumentation grants with the grant no. 441529220 and 445120888 (INST 292/156-1 FUGG (M.L.) and INST 292/154-1 FUGG (M.L.)). Support for the publication fee was provided by University of Greifswald's publication fund. We thank HZB/BESSY in Berlin and EMBL/DESY in Hamburg for continuous support in X-ray data collection.

## Author contributions

M.J. did expression, purification, crystallization of proteins and performed most biochemical experiments. L.B, D.W., and S.S. helped in expression, purification and crystallization of proteins and performed molecular cloning and site-directed mutagenesis. M.J., M.L., G.J.P. collected X-ray data and M.J. and G.J.P. solved the structures. S.S., H.R., and K.G. performed biochemical experiments. N.W. and T.S. supported cloning and growth experiments. L.B., D.W., and B.G. did technical support. I.M. and M.J. performed HPLC analyzes. Ma.D., U.T.B., S.K. performed structure predictions. N.G. performed MD simulations and data evaluation. Mi.D. supervised MD simulations. M.L. initiated, designed, and supervised the study. M.J. and M.L. wrote the manuscript. All authors contributed to data analysis and gave comments on writing the manuscript.

## Funding

## Competing interests

The authors declare no competing interests.
