## [Transparent Peer Review file · Nature Communications]

AcuB senses cellular energy charge to coordinate acetyl-CoA synthesis in bacteria

Corresponding Author: Professor Michael Lammers

Version 0:

Reviewer comments:

Reviewer #1

(Remarks to the Author)

In this manuscript, authors defined AcuB as a new energy sensor, which senses cellular ATP/AMP concentration and then transforms the signal into acetyl-CoA-synthetase AcsA activity that generates acetyl-CoA from acetate. Authors found that AcuB is an adenine nucleotide binding protein and nucleotide binding increases the thermostability of AcuB. AMP-bound AcuB binds AcuC and consequently inhibits the AcsA-deacetylase activity of AcuC. Structural analyses and molecular stimulation partially explains how different adenine nucleotides induce different conformations of AcuB to modulate its binding ability with AcuC, thereby regulating AcsA activity. This study provides the new link in AcsA activity and cellular energy state. However, several issues must be addressed before the work can be considered for publication.

1. One major finding in this study is the nucleotide-binding-dependent AcuB-AcuC interaction. However, authors just used SEC to characterize AcuB-AcuC interaction (Fig 3B). This major conclusion should be reinforced by other complementary biochemical assays (e.g. pull-down or others). Ideally, the quantitative methods to characterize the binding affinity (or kinetics) will provide more insights into the interaction between AcuB and AcuC.

2. How does AcuB inhibit AcuC? Although authors provided biochemical evidence that AcuB inhibits AcuC deacetylase activity, how AcuB inhibits AcuC remains unsolved. Since authors already have the predicted AcuB-AcuC complex model, could authors provide any structural insights into the inhibition mechanism? Any conformational change on AcuC upon AcuB-binding compared with apo AcuC?

3. Authors used AF3 to predict AcuB-AcuC complex structure. Although AF3 is widely used in literatures, the mutagenesis study is usually required to validate the structural model, which is missing in the current manuscript. If authors could design and test some interface mutations based on the predicted model, these data will strengthen the confidence of the complex model.

4. Another AF3-related question. Did authors try to use AF3 to predict complex structure bound with different adenine nucleotide ligands? AF3 has the capacity to do it. It may provide some clues on the potential conformational changes induced by nucleotide binding.

The followings are some minor yet important points.

5. Supplementary Figure 3 shows the elution volumes of different adenine nucleotides. The standard ADP peak is 5.8 ml. However, in most BsAcuB samples, there is a 5.4 ml peak which is assigned as ADP. The elution volume of HPLC is usually very accurate. Is that possible that the 5.4 ml peak does not belong to ADP and it might be another form of adenine nucleotide?

6. Line 204-205, "This indirectly suggests that Ap4A shows the highest binding affinity towards the AcuB proteins of all nucleotides tested". This is an inaccurate statement. How T_m increased on nanoDSF relies on the complex conformation, but not the binding affinity. In the current nanoDSF condition, all AcuB should be saturated by nucleotides.

7. Some typos. For example, line 236, (Fig. 2; Supplementary Fig. 7). It should be Supplementary Fig. 8. Another example, line 376, Fig. 4 should be Fig. 5a.

8. Authors should improve the writing style. The very long paragraph without any break is unfriendly to readers.

Reviewer #2

(Remarks to the Author)

Summary: The manuscript addresses an important question in the regulation of acetyl-CoA synthetase in bacteria by

enzymes encoded by the *acu* operon: What is the function of *AcuB* and how does it achieve it? The authors first show that *AcuB* from three different bacterial species binds adenine nucleotides (AMP, ADP, ATP, and AP4A). They then determine that *AcuB* inhibits *AcuC* in a nucleotide dependent manner with the most potent inhibition attributed to AMP through the formation of a hetero-tetramer complex of the *AcuB* dimer and two *AcuC* molecules (*AcuB*₂:2*AcuC*). The authors have determined crystal structures of *AcuB* without nucleotides and with AMP, AMP-ADP, AMP-ATP, ADP-ATP, AP4A. The authors further carry out Replica Exchange MD simulations on the *AcuB*₂:2*AcuC* hetero-tetramer and the *AcuB*₂ dimer in the presence and absence of nucleotides. Based on the simulation data the authors suggest that the binding of AMP selects the right conformation of *AcuB* which can effectively bind to *AcuC* and inhibit its function.

General comment: This is an interesting study and the authors have put in significant effort to bring out the function of *AcuB*, the unknown member of the *acu*-operon as an inhibitor of *AcuC* activity. However, as noted in my comments below, the data presented on nucleotide binding and functional assays are quite complicated to interpret. Furthermore, I find the conclusions on the mechanistic aspects of the study are not supported by the structural data and appear to be speculative. Therefore I'm unable to recommend the manuscript for publication in nature communications.

Major Comments:

1) The nucleotide binding in *BsAcuB* shows great heterogeneity in loading, stoichiometry and relative ratios of AMP to ADP for different preparations. For *BaAcuB* and *GsAcuB* the max loading is much lower. Overall, AMP/ADP/ATP appear to be rather weak binders whose binding the authors were not able to quantify. Looking at this evidence and the much stronger binding of AP4A should we think of AMP/ADP as the primary ligand for *AcuB*?

2) In the functional assays, the inhibition of *AcuC* with *AcuB* is accomplished in a nucleotide independent manner (for *BsAcuB* and *GsAcuB*), which the authors attribute to the minority AMP bound population. But this is not proven as the Apo forms of *AcuB* were not available. Further the authors show that the ACT domain by itself can inhibit *AcuC* which is also independent of AMP. Doesn't this imply that even Apo *AcuB* should be able to inhibit *AcuC*?

3) In the biological context, nucleotides are expected to bind divalent cations (Mg²⁺, Mn²⁺). However, the crystal structures do not show any bound ions. Is this because of the low resolution of the crystal structures or were these ions not present in the crystallization conditions? This factor may be important in the discussion of electrostatics in the different nucleotide bound states and also could be a consideration for the MD simulations.

4) The authors indicate that conformational flexibility as an important factor which regulates *AcuB* function. However, as the authors themselves acknowledge there is no evidence from crystal structures that the binding of different nucleotides or the absence of nucleotides creates major structural changes in *AcuB*.

5) In the MD simulations, flat bottomed RMSD constraints are applied which are suitable in biased sampling simulations such as metadynamics etc but not recommended in a method like Replica Exchange which accelerates the sampling of the energy landscape. Applying constraints beats the purpose of REX which is aimed at enhancing the extent of energy landscape sampled. In fact, in the absence of constraints it would be better to assess if the complex becomes unstable and dissociates in the absence of nucleotides/ADP/ATP/AP4p. Rather than RMSD, I would suggest using geometric distances/angles between domain COMs and energetics to assess weakening of protein-protein interactions.

Minor comments:

1) In the abstract, no motivation is provided for studying *AcuB* (the preceding sentences focus completely on *AcuA* and *AcuC* regulating the activity of *AcsA*). Even in the Introduction, very little of the background/challenges in lack of understanding for *AcuB* is provided. It would be good to explicitly cover this both in Abstract and Introduction to orient the broader readerbase.

2) Please run basic spell/grammar checks on the entire MS. See for e.g pg 7 line 197 "...molar access..", line 199 "...similar extend.", 209 "...analysed which role the presence...". On Pg 9 line 270 "..*BsAcuC* with *BsAcuC*..". Also I found highlighted text which were remnants of revisions carried out by the authors.

Reviewer #3

(Remarks to the Author)

Acetate metabolism is important to maintain the cellular energy balance in bacteria. The consumption and accumulation of acetate were precisely-regulated in bacteria to adapt the changed environment especially the available of carbon sources. In this work, the authors found that the *AcuB*, encoded by *acu*-operon, could reversibly bind *AcuC* in different cellular AMP level, thus resulting in inactivation or activation of *AcuC*. The *AcuC* is a deacetylase functioned in regulate the activity of AMP-forming acetyl-CoA-synthetase *AcsA*. This finding filled an important gap in regulation of *AcsA* activity and the coordination of acetate assimilation and dissimilation. The manuscript is well conceptualized and the data are presented in a logical manner. I have a few comments for the authors to address:

1. This work focused on the *acu*-operon in gram-positive Bacilli. Is *acu*-operon or *acu*-like operon existed in gram-negative bacteria such as *Escherichia coli*? Whether *AcuB* is a widespread energy sensor in regulating acetate assimilation and

dissimilation in bacteria?

2. From the aspect of biotechnology, is AcuB a potential bioengineering target for enhancing the acetate assimilation in bacteria?

3. I suggest divide some long paragraphs into shorter ones.

4. I suggest reorganize Fig. 8 for better coordinating the regulation lines, energy flows and acetate metabolism pathways.

Version 1:

Reviewer comments:

Reviewer #1

(Remarks to the Author)

Most of my previous comments have been addressed, except for points 1 and 8. Regarding the characterization of the nucleotide-dependent interaction between AcuB and AcuC, the authors have included additional data such as nanoDSF experiments and the Fluor de Lys assay. However, these newly added data do not directly measure protein-protein interaction. Rather, they reflect changes in protein conformation, not the actual binding affinity. Since a key conclusion of this manuscript is that AMP enhances the interaction between AcuB and AcuC, a quantitative method to assess the binding affinity between the two proteins would be preferable. Such an approach could directly demonstrate the extent to which different nucleotides modulate their interaction. If this is not feasible, the authors should explicitly discuss the limitations of the current study.

In my original comments, I also noted that "the excessively long paragraph without breaks is unfriendly to readers." Unfortunately, this issue remains in the revised version. The Discussion section consists of a single paragraph spanning six pages, and the Introduction similarly contains an unbroken paragraph that extends over four pages.

Reviewer #2

(Remarks to the Author)

The authors have extensively revised the manuscripts adding additional experiments/data (nanoDSF, mutational analysis, additional AlphaFold models) and also clarified the application of restraints in their MD protocol. With these changes, my concerns are largely addressed. I'm happy to recommend the manuscript for publication.

Reviewer #3

(Remarks to the Author)

The authors have addressed my comments reasonably well. I do not have further comments.

Version 2:

Reviewer comments:

Reviewer #1

(Remarks to the Author)

The authors have addressed my comments, so now I support the publication of this manuscript.

Point-by-point response to the reviewers' comments

Reply to all reviewers:

We thank the reviewers for carefully reading our manuscript and giving us these valuable and constructive suggestions. We integrated these suggestions into the revised manuscript from our point of view strongly improving the manuscript. We worked comprehensively on all open points. As suggested by the reviewers, we added experimental data **showing complex formation of AcuB and AcuC by analytical size-exclusion chromatography (SEC), by nanoDSF experiments, enzyme kinetics (Fluor-de-Lys-assay and immunoblotting)**. These additional data support, that addition of AMP drives complex formation. We additionally prepared **several AcuB mutants** that allowed us to confirm the AlphaFold3 model of the heterotetrameric AcuB₂•2AcuC complex. The mutants were analyzed by **analytical SEC and HPLC** confirming that they behaved similar as AcuB wildtype eluting as dimer and showing a similar nucleotide loading. The influence of these mutants on **complex formation with AcuC was studied by analytical SEC**. We discovered that a mutation in the ACT domain and a mutation in the Bateman domain of AcuB affect the interaction with AcuC supporting the AlphaFold3 model showing AcuC contacts both domains. Moreover, we **determine the inhibitory constant, IC₅₀, for the inhibition of AcuC deacetylase activity by AcuB**. This data allows us to **quantify the binding affinity** for the interaction of AcuB and AcuC in the presence of AMP to reside in the nanomolar range and confirming the AcuB mutants affect binding. We conducted further **AlphaFold3 structure predictions to assess how AcuB binding to different adenine nucleotides affects its conformation**. These data reveal a highly similar overall conformation of AcuB independent of the bound type of adenine nucleotide. However, our data also suggests that AlphaFold3 is not able to correctly model the conformational dynamics occurring within the Bateman domain upon adenine nucleotide binding. We worked on the **language and the writing style** of the manuscript as suggested by the reviewers and subdivided the **text in smaller paragraphs** when it was reasonable.

Reviewer #1 (Remarks to the Author):

In this manuscript, authors defined AcuB as a new energy sensor, which senses cellular ATP/AMP concentration and then transforms the signal into acetyl-CoA-synthetase AcsA activity that generates acetyl-CoA from acetate. Authors found that AcuB is an adenine nucleotide binding protein and nucleotide binding increases the thermostability of AcuB. AMP-bound AcuB binds AcuC and consequently inhibits the AcsA-deacetylase activity of AcuC. Structural analyses and molecular stimulation partially explain how different adenine nucleotides induce different conformations of AcuB to modulate its binding ability with AcuC, thereby regulating AcsA activity. This study provides the new link in AcsA activity and cellular energy state. However, several issues must be addressed before the work can be considered for publication.

Point 1:

One major finding in this study is the nucleotide-binding-dependent AcuB-AcuC interaction. However, authors just used SEC to characterize AcuB-AcuC interaction (Fig 3B). This major conclusion should be reinforced by other complementary biochemical assays (e.g. pull-down or others). Ideally, the quantitative methods to characterize the binding affinity (or kinetics) will provide more insights into the interaction between AcuB and AcuC.

Response:

Our analytical size-exclusion chromatography (SEC) data clearly shows that AcuB forms a complex with AcuC, as the proteins co-elute from the column and that this interaction is dependent on the presence of AMP. These analytical SEC experiments were performed using AcuC and AcuB in the micromolar range suggesting that the affinity is higher than micromolar. Along that line, upon addition of AMP we observe a complex peak representing an 2AcuC•AcuB₂ complex at these concentration suggesting an affinity in the (sub-)micromolar range. For the proteins GsAcuB and GsAcuC the elution peaks quantitatively shifted upon addition of AMP indicating the quantitative formation of the complex as no major pool of free GsAcuC and GsAcuB is detectable. Besides, other methods that clearly show the interaction between AcuB and AcuC are the enzyme assays using either a fluorescence readout and an AMC-labeled Boc-Lys(Ac)-AMC model substrate or using a deacetylation of the physiological substrate of AcuC, the AMP-forming acetyl-CoA synthetase (AcsA), and immunoblotting as a readout. To be able to more quantitatively characterize the interaction we performed inhibition assays. We determined dose-response curves assessing the inhibitory potency of AcuB on AcuC deacetylase activity that allowed us to determine IC₅₀ values showing the potency of AcuB to inhibit AcuC in presence of AMP. This does directly correlate with the K_D-value for the interaction of AcuB and AcuC as the binding of AcuB determines its inhibitory potency. These data show that AcuB is able to inhibit AcuC with an IC₅₀ value of app. 38 nM in the presence of AMP, while addition of other nucleotides, i.e. ADP, ATP and Ap4A, do not allow complex formation. We also analyzed AcuB-mutants, i.e. GsAcuB M189R in the ACT domain and H87W in the Bateman domain, to validate the AlphaFold3 model. These data are included in the revised manuscript (as Fig. 7a,b). To furthermore confirm the interaction of GsAcuB and GsAcuC we performed additional experiments and analyzed complex formation of GsAcuB and GsAcuC by nanoDSF

measurements. These data suggest that AcuC alone does not bind to any of the adenine nucleotides analyzed (Supplementary
Fig. 11). Moreover, addition of all nucleotides result in stabilization of the GsAcuB protein compared to the sample without
addition of nucleotides. However, all nucleotides vary in their stabilizing effect (Fig. 1d; Supplementary Fig. 11). Notably, also
nanoDSF data of GsAcuB, GsAcuC and their mixture show, that in contrast to the sample without addition of exogenous
nucleotides, for all samples with nucleotides the fluorescence of the mixture is not a linear combination of its components. This
proofs interaction of GsAcuB with GsAcuC, most likely complex formation occurs upon addition of all adenine nucleotides to
some extent (Supplementary Fig. 11). Overall, these data suggest that binding of AMP to AcuB stabilizes a conformation
which is most competent to bind to and inhibit AcuC-deacetylase activity ultimately translating into a sustained inactivation of
AcsA activity under conditions of low intracellular energy charge. We hypothesize that at least one phosphate is needed per
nucleotide binding site acting as counterions to neutralize the positive charges in the Bateman domains otherwise experiencing
electrostatic repulsion. In the nucleotide free state AcuB is not capable to bind to AcuC, i.e. the positive charges need to be
neutralized for AcuB to adopt a conformation that is compatible with AcuC binding. However, inserting adenine nucleotides
with more phosphate groups, i.e. ADP and/or ATP, would result in insertion of negative charges far more than needed for
neutralization of the positive charges and adds a steric component overall resulting in a conformation that disfavors AcuC
binding compared to AMP. According to the conformational selection theory we propose that AcuB can exist in multiple
conformations but binding of AMP stabilizes a conformation that is most appropriate for AcuC binding and as a consequence
inhibition of AcuC. As a summary, our experimental, structural and MD simulations data clearly show that AcuB is capable
to inhibit AcuC most potently in presence of AMP but neither in its nucleotide free state nor in presence of ADP, ATP or
Ap4A.

**Point 2:**

How does AcuB inhibit AcuC? Although authors provided biochemical evidence that AcuB inhibits AcuC deacetylase
activity, how AcuB inhibits AcuC remains unsolved. Since authors already have the predicted AcuB-AcuC complex
model, could authors provide any structural insights into the inhibition mechanism? Any conformational change on
AcuC upon AcuB-binding compared with apo AcuC?

**Response:**

We thank the reviewer for raising this point. The analytical size-exclusion chromatography experiments suggest that
AcuB and AcuC form a heterotetrameric complex consisting of two molecules of AcuB and two molecules of AcuC. We
performed AlphaFold3 structure predictions of the complex between AcuB and AcuC. These models suggest the
formation of a 2AcuC•AcuB₂-complex, i.e., a dimer of AcuB binds to two molecules of AcuC. The major contact of each
AcuC molecule towards AcuB is formed with the C-terminal ACT domain of each AcuB monomer, however, each AcuC
molecule also contacts the Bateman domain of the other AcuB monomer. This allows AcuC to sense conformational
changes within the AcuB dimer occurring upon nucleotide binding. The predicted AlphaFold3 models have robust quality
indicators, including pTM+ipTM-values, suggesting the validity of the models (Supp. Table 7). The AlphaFold3 model
suggests that AcuB binds to AcuC in a way restricting substrate access to the AcuC active site. To visualize this blockage
of the active site entry in AcuC by AcuB binding, we prepared an AlphaFold3 model of a K549-acetylated AcsA peptide
in complex with AcuC and superposed this with the predicted structure of the 2AcuC•AcuB₂-complex (Supp. Fig. 31). So
far, no experimental structure is available for AcuC alone. Earlier data performed in our lab shows that AcuC forms a
monomer in solution (Graf, L.G. et al. Nat Commun 15, 9496 (2024). <https://doi.org/10.1038/s41467-024-53903-0>).
Inspecting the AcuC AlphaFold3 model and the 2AcuC•AcuB₂-complex predicted by AlphaFold3 reveals no significant
conformational changes occurring in AcuC upon binding towards AcuB.

**Point 3:**

Authors used AF3 to predict AcuB-AcuC complex structure. Although AF3 is widely used in literatures, the mutagenesis
study is usually required to validate the structural model, which is missing in the current manuscript. If authors could
design and test some interface mutations based on the predicted model, these data will strengthen the confidence of the
complex model.

**Response:**

We agree with the reviewer and designed two mutants in AcuB to confirm the AlphaFold3 model of the complex between
AcuB and AcuC. Concerning the mutations, we selected one residue within the AcuB ACT domain, i.e. M189, and one
residue in the AcuB Bateman domain, i.e. H87, to confirm the contacts of AcuC to the ACT domain and the Bateman
domain of AcuB. The M189 was mutated to Arg, i.e. M189R, and H87 to Trp, i.e. H87W. M189 is solvent-exposed in the
non-complexed form of AcuB and it penetrates deeply into the AcuC active site upon forming the AcuB-AcuC-complex.
We selected the mutation of M189 to R since the positively-charged Arg will sterically and electrostatically interfere with
AcuC binding while not interfering with the solubility of AcuB. H87 located in the Bateman domain of AcuB is in interaction
distance to AcuC and mutation to the bulky Trp at this site will sterically and electrostatically interfere with AcuC binding
while not interfering with solubility of AcuB. We highlighted both residues in Supp. Fig. 29. We performed analytical size-
exclusion chromatography experiments to show that both mutants, i.e. AcuB M189R and AcuB H87W, behaved like

AcuB wildtype eluting as apparent dimer (Supp. Fig. 30; Supp. Table 8). Moreover, we assessed the nucleotide loading
by HPLC and observed that both mutants are loaded mostly to AMP after expression in *E. coli* and subsequent
purification of the proteins suggesting that these mutations do not interfere with nucleotide binding (Supp. Fig. 3).
Analytical SEC-experiments show that AcuB M189R is not capable to form a complex with AcuC in presence of AMP
confirming the importance of M189 for the interaction with AcuC. For AcuB H87W we still observed complex formation
with AcuC in presence of AMP, however, this was strongly impaired compared to AcuB wildtype. To confirm these data,
we performed an additional fluorescence Fluor-de-Lys assay using Boc-Lys(Ac)-AMC as substrate to assess the AcuC
deacetylase activity in presence of AcuB wildtype, AcuB M189R or AcuB H87W. These data show that AcuB wildtype is
able to inhibit AcuC activity in presence of AMP (IC₅₀: 38 nM) while AcuB M189R almost completely lost its capacity to
inhibit AcuC and AcuB H87W shows a mild inhibitory effect on AcuC activity (IC₅₀: 71 nM; Fig. 7b,c). Overall, these data
confirm the AlphaFold3 model of the complex between AcuB and AcuC showing, the AcuB contacts AcuC with the ACT
domain and the Bateman domain with major contacts to the ACT domain. Moreover, the activity assays also confirm the
AlphaFold3 model showing that binding of AcuB to AcuC inactivates AcuC activity by blocking AcuC-active site access
of the deacetylation-substrate. These data were included in the revised manuscript.

**Point 4:**
Another AF3-related question. Did authors try to use AF3 to predict complex structure bound with different adenine
nucleotide ligands? AF3 has the capacity to do it. It may provide some clues on the potential conformational changes
induced by nucleotide binding.

**Response:**

We included AlphaFold3 models for *B. subtilis* and *G. stearothermophilus* AcuB in their complexes with various adenine
nucleotides and variations thereof (Supplementary Figs. 32,33). Using these models we performed structural alignments
on the experimental structures of either BsAcuB determined in complex with 2AMP and 2 ADP or GsAcuB determined
in the nucleotide-free, empty state. The alignments were done on the Bateman domains of monomer A (residues 1-131).
Overall, the nucleotide-bound models show a high structural similarity (Supplementary Table 9). However, there are
differences in the experimentally determined structures/AlphaFold3 models of nucleotide-free AcuB-proteins compared
to the nucleotide-bound structures. These differences are obvious in the Bateman domains as well as the ACT domains
as indicated by C α -atom distances of selected residues (Supplementary Fig. 32,33). As indicated for the AlphaFold3
BsAcuB models, these models might not represent the real protein dynamics. AlphaFold3 modeled two ATP-molecules
in each chain A and chain B in the GsAcuB•4ATP complex with the consequence that the γ -phosphates distances are
only 1.9 Å. In the BsAcuB•4ATP model even a covalent bond between the γ -phosphates of two ATP-molecules were
predicted. The GsAcuB model in complex with two Ap4A molecules shows each Ap4A molecule binds only within one
GsAcuB monomer rather than bridging both monomers in the dimer as occurring in the experimental structure of
GsAcuB•2Ap4A. We highlighted conformational differences accessible by AlphaFold3 structure predictions within the
Bateman domains as indicated by the distance of the C α -atoms of Leu45 of 7.2 Å and in the ACT domains indicated by
the distance of C α Lys153 of 7.0 Å and Glu160 of 7.5 Å (Supplementary Fig. 32,33). This suggests that nucleotide-
binding affects the conformation of GsAcuB which is also experimentally supported by the nanoDSF data. Overall,
AlphaFold3 is not able to accurately model the dynamics in AcuB occurring upon binding to different nucleotides. The
binding of different adenine-nucleotides with different numbers of phosphates leads to additional electrostatic and steric
components within the models. Binding of different adenine nucleotides with a varying number of phosphate groups
affects the conformation of the Bateman domain resulting in a more open or closed conformation. Our structural data
shows that binding of the adenine nucleotides at both nucleotide binding sites per AcuB monomer is predominantly
mediated by binding the adenine-base, ribose-moiety and the α -phosphate. The presence of additional phosphates in
ADP and/or ATP results in steric and electrostatic effects influencing the conformation of the AcuB Bateman domains
within the dimer. This in turn modulates the affinity of AcuC binding towards AcuB. AlphaFold3 is able to predict structures
based on the data in the PDB, with which it was trained with. This does not account for structural dynamics occurring
upon binding of ligands with differences in their steric and electrostatic components. Particularly, electrostatic effects by
repulsion of positively charged residues in the Bateman domains in the nucleotide free AcuB or negatively charged
adenine-nucleotide phosphates act over long-distances affecting the dynamics in the Bateman domains. The presented
data, including MD simulations and enzyme assays, clearly show differences in the AcuB-conformations occurring upon
binding to different adenine nucleotides. Only in presence of AMP we do observe a strong inhibitory potential of AcuB
on AcuC as shown by different methodological approaches, i.e. MD simulations, enzyme assays, nanoDSF- and
analytical SEC-experiments. We added the novel experimental data and those for the AlphaFold3 models of AcuB in
complexes with various nucleotides to the revised manuscript.

**Point 5:**

Supplementary Figure 3 shows the elution volumes of different adenine nucleotides. The standard ADP peak is 5.8 ml.
However, in most BsAcuB samples, there is a 5.4 ml peak which is assigned as ADP. The elution volume of HPLC is
usually very accurate. Is that possible that the 5.4 ml peak does not belong to ADP and it might be another form of
adenine nucleotide?

**Response:**

The reviewer is correct. These inconsistencies were due to the fact that for these protein preparations the HPLC-runs
were conducted with a HPLC-valve mixing solutions of acetonitrile and TBAB, for others an isocratic elution mode was
performed that had acetonitrile and TBAB present in one buffer solution. However, we can confirm that the results are
correct as stated in the manuscript and the peaks labelled with ADP are ADP-peaks. We performed isocratic elution for
all following HPLC-runs to avoid the problems obtained with the mixing valve.

**Point 6:**
Line 204-205, "This indirectly suggests that Ap4A shows the highest binding affinity towards the AcuB proteins of all
nucleotides tested". This is an inaccurate statement. How T_m increased on nanoDSF upon Ap4A relies on the complex
conformation, but not the binding affinity. In the current nanoDSF condition, all AcuB should be saturated by nucleotides.

**Response:**
We thank the reviewer for this point. Reviewer 1 is correct. We cannot conclude that Ap4A shows the highest affinity
towards AcuB compared to the other nucleotides using nanoDSF. For these experiments, concentrations of 1 mM of
nucleotides were used and 1 $\mu\text{g}/\mu\text{L}$ (33.3 μM) of AcuB protein so that all nucleotides were used at saturating
concentrations. Rather Ap4A stabilizes the conformation of the AcuB protein the most compared to the other tested
nucleotides. This is most likely due to the fact that Ap4A is able to intermolecularly bridge the two AcuB monomers of
the dimer by binding to the opposite nucleotide binding site of the two AcuB molecules within the dimer. We removed the
wrongly stated sentence to be scientifically precise.

**Point 7:**
Some typos. For example, line 236, (Fig. 2; Supplementary Fig. 7). It should be Supplementary Fig. 8. Another example,
line 376, Fig. 4 should be Fig. 5a.

**Response:**
We corrected the typos as suggested by the reviewer.

**Point 8:**
Authors should improve the writing style. The very long paragraph without any break is unfriendly to readers.

**Response:**
We worked on language and writing style. We subdivided the paragraph entitled „AcuB adopts different conformational
states dependent on the nucleotide-loading state“ into two paragraphs adding the new experimental data and AlphaFold3
models of AcuB in complexes with various adenine nucleotides in the paragraph entitled „AcuB and AcuC form a
heterotetrameric AcuB₂•2AcuC-complex“ to improve readability.

**Reviewer #2 (Remarks to the Author):**

Summary: The manuscript addresses an important question in the regulation of acetyl-CoA synthetase in bacteria by
enzymes encoded by the acu operon: What is the function of AcuB and how does it achieve it? the authors first show
that AcuB from three different bacterial species binds adenine nucleotides (AMP,ADP, ATP, and AP4A). They then
determine that AcuB inhibits AcuC in a nucleotide dependent manner with the most potent inhibition attributed to AMP
through the formation of a hetero-tetramer complex of the AcuB dimer and two AcuC molecules (AcuB₂:2AcuC). The
authors have determined crystal structures of AcuB without nucleotides and with AMP, AMP-ADP, AMP-ATP, ADP-ATP,
AP4A. The authors further carry out Replica Exchange MD simulations on the AcuB₂:2AcuC hetero-tetramer and the
AcuB₂ dimer in the presence and absence of nucleotides. Based on the simulation data the authors suggest that the
binding of AMP selects the right conformation of AcuB which can effectively bind to AcuC and inhibit its function.

General comment: This is an interesting study and the authors have put in significant effort to bring out the function of
AcuB, the unknown member of the acu-operon as an inhibitor of AcuC activity. However, as noted in my comments
below, the data presented on nucleotide binding and functional assays are quite complicated to interpret. Furthermore, I
find the conclusions on the mechanistic aspects of the study are not supported by the structural data and appear to be
speculative. Therefore I'm unable to recommend the manuscript for publication in nature communications.

**Point 1:**
The nucleotide binding in BsAcuB shows great heterogeneity in loading, stoichiometry and relative ratios of AMP to ADP
for different preparations. For BaAcuB and GsAcuB the max loading is much lower. Overall, AMP/ADP/ATP appear to
be rather weak binders whose binding the authors were not able to quantify. Looking at this evidence and the much
stronger binding of Ap4A should we think of AMP/ADP as the primary ligand for AcuB?

**Response:**
The reviewer is correct in saying that this system is rather complex, which makes it difficult to study. However, we think
we have robust data supporting our model. From our data we can conclude that the nucleotide affinity is of medium
affinity. Although we were not able to quantify the affinity of AcuB to the nucleotides due to the difficulty to prepare fully
nucleotide free AcuB-protein and as nucleotide-free AcuB is quite instable, we conclude this medium affinity from the
fact that to some extent nucleotides co-elute during purification of AcuB. Therefore, we expect the affinity to be in the

micromolar to sub-micromolar range. Importantly, nanoDSF data indicate that Ap4A has the highest stabilizing effect on
AcuB as visible by the highest T_m obtained upon addition of Ap4A compared to other nucleotides analyzed. For these
experiments, concentrations of 1 mM of nucleotides were used and 1 $\mu\text{g}/\mu\text{L}$ (33.3 μM) of AcuB so that all nucleotides
were used at saturating concentrations when considering an affinity in the low micromolar range. Rather Ap4A stabilizes
the conformation of the AcuB protein resulting in stabilization of the protein and as a consequence to the strong increase
in the melting temperature. Our structural data supports this observation as Ap4A is able to intermolecularly bridge the
two AcuB monomers of the dimer by binding two nucleotide binding sites on the same face of the dimer. Our data also
clearly show that AcuB binds to AMP nucleotides and the structures presented here also indicate that in principle AcuB
can also bind to ADP and ATP. Most contacts are realized by the adenine base, the ribose sugar and the α -phosphate
for all nucleotides bound and the additional phosphate groups in ADP and/or ATP play a role on the AcuB-conformation
exerting electrostatic and steric effects. To this end, we conclude that AcuB does not only play a role under cellular stress
conditions with Ap4A being important but does play a role to adjust the AcuC activity also under physiological and non-
stress conditions depending on which adenine nucleotide species dominates under different growth conditions. However,
under stress-conditions in which Ap4A-concentrations increase, Ap4A binding to AcuB might surely also play a role in
inhibiting AcuB binding to AcuC thereby indirectly ensuring that acetyl-CoA synthesis can take place. For these
discussions, the concentrations of different nucleotides play an important role. The intracellular concentrations for Ap4A
were reported to be in the range of 20 μM under non-stress conditions, while the intracellular concentrations of
AMP/ADP/ATP are reported to be much higher. To this end, Ap4A might likely be outcompeted by other nucleotides
under non-stress conditions. Overall, our data shows that AcuB does play a role as a sensor for different adenine-
nucleotides under normal physiological conditions not excluding to play a role also under stress conditions upon increase
in cellular Ap4A concentration.

**Point 2:**

In the functional assays, the inhibition of AcuC with AcuB is accomplished in a nucleotide independent manner (for
BsAcuB and GsAcuB), which the authors attribute to the minority AMP bound population. But this is not proven as the
Apo forms of AcuB were not available. Further the authors show that the ACT domain by itself can inhibit AcuC which
is also independent of AMP. Doesn't this imply that even Apo AcuB should be able to inhibit AcuC?

**Response:**

Our data clearly shows that AcuB can potentially bind and inhibit AcuC in the presence of AMP, whereas nucleotide free
AcuB does neither bind to AcuC nor does it inhibit AcuC. Our experimental data regarding the inhibition of AcuC activity
in the Fluor-de-Lys assay assessing deacetylation of a model substrate by AcuC and in the assay assessing the capacity
of AcuC to deacetylate AcsA indicate that the isolated ACT domain is to some extent capable to inhibit AcuC. However,
the inhibitory potency of the isolated ACT domain is significantly lower compared to the full AcuB protein in the presence
of AMP. Along that line, the analytical size-exclusion chromatography data shows that the ACT domain does not form a
stable complex with AcuC, while full length AcuB in presence of AMP does. The AlphaFold3 model suggests that in the
complex with AcuB dimer AcuC contacts the ACT domain of one monomer and the Bateman domain of the other
monomer. We confirmed the AlphaFold3 model with mutational analyses and included this data in the revised
manuscript. To this end, we constructed the AcuB mutant M189R that is located within the ACT domain and the mutant
H87W located within the Bateman domain. Both AcuB mutants show a reduced complex formation in presence of AMP
and impaired the inhibitory potency on AcuC. While AcuB inhibited AcuC with an IC_{50} of 38 nM, AcuB M189R did not
inhibit AcuC and AcuB H87W shows a reduced IC_{50} of 71 nM (Fig. 7b,c). Addition of other nucleotides, i.e., ADP, ATP
or Ap4A, did not result in an AcuB capable to inhibit AcuC (Fig. 7b,c). See also our answer to Point 3 of reviewer 1.
These data suggest, that on the one hand AcuC has a higher affinity towards the full length AcuB dimer in presence of
AMP as it contacts both, the ACT domain and the Bateman domain. On the other hand, presence of other nucleotides,
i.e. ADP and/or ATP, stabilizes a conformation of AcuB which disfavours binding to AcuC to the AcuB dimer. Overall,
our data shows that the formation of the scissor-like AcuB-dimer allows to sense the nucleotide-state in the cell and
translate this information into its capacity to bind and to inhibit AcuC. This would be impossible with the isolated ACT
domain.

To furthermore confirm the interaction of GsAcuB and Gs AcuC we performed additional experiments and analyzed complex
formation of GsAcuB and GsAcuC by nanoDSF measurements. These data suggest that AcuC alone does not bind to any of
the adenine nucleotides analyzed (Supplementary Fig. 11). Moreover, addition of all nucleotides result in stabilization of the
GsAcuB protein compared to the sample without addition of nucleotides. However, all nucleotides vary in their stabilizing effect
(Fig. 1d; Supplementary Fig. 11). Notably, also nanoDSF data of GsAcuB, GsAcuC and their mixture show that in contrast to
the sample without addition of nucleotides for all samples with nucleotides the fluorescence of the mixture is not a linear
combination of its components. This proves interaction of GsAcuB with GsAcuC, most likely complex formation occurs upon
addition of all adenine nucleotides to some extent (Supplementary Fig. 11). Overall, these data suggest that binding of AMP
to AcuB stabilizes a conformation which is most competent to bind to and inhibit AcuC-deacetylase activity ultimately translating
into a sustained inactivation of AcsA activity under conditions of low intracellular energy charge. We hypothesize that at least

one phosphate is needed per nucleotide binding site acting as counterions to neutralize the positive charges in the Bateman
domains otherwise experiencing electrostatic repulsion. The analytical SEC data, enzyme kinetics and nanoDSF data show i
the nucleotide free state AcuB is not capable to bind to AcuC, i.e. the positive charges need to be neutralized for AcuB to adopt
a conformation that is compatible with AcuC binding. However, inserting adenine nucleotides with more phosphate groups, i.e.
ADP and/or ATP, would result in insertion of negative charges far more than needed for neutralization of the positive charges
and adds a steric component overall resulting in a conformation that disfavors AcuC binding compared to AMP. According to
the conformational selection theory we propose that AcuB can exist in multiple conformations but binding of AMP stabilizes a
conformation that is most appropriate for AcuC binding and as a consequence inhibition of AcuC. As a summary, our
experimental, structural and MD simulations data clearly show that AcuB is capable to inhibit AcuC most potently in
presence of AMP but neither in its nucleotide free state nor in presence of ADP, ATP or Ap4A.

**Point 3:**

In the biological context, nucleotides are expected to bind divalent cations (Mg^{2+} , Mn^{2+}). However, the crystal
structures do not show any bound ions. Is this because of the low resolution of the crystal structures or were these ions
not present in the crystallization conditions? This factor may be important in the discussion of electrostatics in the
different nucleotide bound states and also could be a consideration for the MD simulations.

**Response:**

The reviewer is correct saying that many nucleotide binding proteins use Mg^{2+} (or Mn^{2+}) for nucleotide binding, however,
these proteins either use those metal ions to increase nucleotide affinity or are enzymes using them for catalysis as
represented by many DNA-modifying enzymes. For AcuB to function in a physiological sense a not too high nucleotide
affinity is important as otherwise it could not fulfill its role as a cellular energy sensor. Moreover, AcuB does not have any
enzymatic activity on its own. Instead, it uses binding of different nucleotides to adjust its conformation. We want to stress
that the structures presented here are not of low resolution. For *G. stearothermophilus* we obtained a resolution of 2 Å
and the protein was purified in buffer with 5 mM $MgCl_2$. The crystallization setup was mixed in a protein to crystallization
mother liquid ratio of 1:1 giving a final $MgCl_2$ concentration of app. 2.5 mM in the condition. The resolution is reasonable
high to be able to clearly identify bound Mg^{2+} ions if there were any. In AcuB, there are many positively charged residues
in the nucleotide binding area, including Arg and Lys side chains, within the Bateman domains functionally exerting a
similar role as Mg^{2+} ions to facilitate nucleotide binding by contacting the negatively charged nucleotide phosphate
groups.

**Point 4:**

The authors indicate that conformational flexibility as an important factor which regulates AcuB function. However, as
the authors themselves acknowledge there is no evidence from crystal structures that the binding of different
nucleotides or the absence of nucleotides creates major structural changes in AcuB.

**Response:**

It is a reported feature for Bateman domains that conformational changes are not fully accessible by structures obtained
by X-ray crystallography. This is not restricted to the structures presented here but for other systems as well. It is a well-
known feature for Bateman domains that these can exist in open or closed conformations or even in bent conformations
depending on the ligands bound. Moreover, it is reported that CBS-proteins undergo conformational changes of variable
magnitude upon binding of adenosine derivatives (Ereño-Orbea et al. (2013) doi.org/10.1016/j.abb.2013.10.008). The
crystal structures obtained for nucleotide-bound GsAcuB were obtained by soaking crystals of nucleotide-free GsAcuB.
In these structures the adenine-base, the ribose sugar and the α -phosphates of the nucleotides are well defined by the
electron density. However, for the β -phosphates and even more relevant for the γ -phosphates, we observed only weak
electron density. In fact, the γ -phosphates for ATP were almost not defined. This supports our model that binding of the
adenine-ribose- α -phosphate moiety is very similar to the conformation of nucleotide-free GsAcuB. However, addition of
further phosphates in ADP and/or ATP would induce a different, more extended conformation in the Bateman domains
to be better able to bind the β -/ γ -phosphates. This becomes obvious by the MD-simulations. We made another important
observation that supports this notion. When soaking crystals of BsAcuB with Ap4A, we observed that crystals completely
dissolved suggesting that this induced a conformational change in AcuB not compatible with the actual crystal contacts
of the crystals. To this end, we confirmed the conformational flexibility in the Bateman domains by additional MD
simulations showing that the conformations in the AcuB-Bateman domain differs depending on the nucleotide-loading
state by the bound nucleotides exerting an electrostatic and steric effect with the negatively-charged phosphate groups.
From our perspective, our data is strong enough to support this model.

**Point 5:**

In the MD simulations, flat bottomed RMSD constraints are applied which are suitable in biased sampling simulations
such as metadynamics etc but not recommended in a method like Replica Exchange which accelerates the sampling

of the energy landscape. Applying constraints beats the purpose of REX which is aimed at enhancing the extent of
energy landscape sampled. In fact, in the absence of constraints it would be better to assess if the complex becomes
unstable and dissociates in the absence of nucleotides/ADP/ATP/Ap4p. Rather than RMSD, I would suggest using
geometric distances/angles between domain COMs and energetics to assess weakening of protein-protein
interactions.

**Response:**

This concern would apply if RMSD restraints had been imposed on the AcuB–AcuC complex as a whole. However, in
our simulations, the flat-bottomed RMSD restraints were applied separately to each individual protein (each AcuB and
each AcuC), not to the complex. RMSD deviations for these restraints also came with prior position and orientation
alignment and hence neither restricted position, nor orientation. Consequently, the relative motions between AcuB and
AcuC, including potential weakening of interactions or dissociation events, were not artificially restricted and could be
fully sampled within the replica exchange framework. Moreover, the restraints were defined with a relatively permissive
threshold of 8 Å and were introduced solely to prevent complete unfolding of the individual proteins at elevated
temperatures, rather than to bias inter-protein interactions. We acknowledge that this distinction was not described with
sufficient clarity in the original Methods section and have therefore revised the manuscript to explicitly state that the
restraints were applied independently to each AcuB and each AcuC. Finally, given the observed rolling and sliding
motions of AcuC on the surface of AcuB, we consider RMSD to be an appropriate metric for assessing the overall stability
of the complex in our system.

**Point 6:**

In the abstract, no motivation is provided for studying AcuB (the preceding sentences focus completely on AcuA and
AcuC regulating the activity of AcsA. Even in the Introduction, very little of the background/challenges in lack of
understanding for ACuB is provided. It would be good to explicitly cover this both in Abstract and Introduction to orient
the broader readerbase.

**Response:**

We modified the abstract and the introduction as suggested by the reviewer and think that the gap-of-knowledge is
clearer now. So far, little is known about AcuB in bacteria and the function of Bateman domain containing proteins in
bacteria. In eukaryotes, the formation of multi-protein complexes was shown to regulate the activity and determine the
substrate specificity of classical Zn²⁺-dependent deacetylases. This is completely unknown in prokaryotes and in fact, to
our knowledge, this is the first example of a protein regulating the activity of a classical deacetylase, i.e. AcuC, in bacteria.
Along that line, it is also not understood how the counteracting activities of AcuA, acting as acetyltransferase for AcsA,
and AcuC, acting as deacetylase for AcsA, are regulated. AcuB is a major player coordinating these activities.

We rewrote the abstract and think that the gap-of-knowledge becomes clear:

„...How the counteracting activities of AcuA and AcuC are regulated is not understood. Here, we close this gap and performed
a structure-function analyses on AcuB. This reveals that AcuB forms a scissor-shaped dimer with each monomer consisting of
an N-terminal Bateman domain binding to adenine nucleotides and a C-terminal ACT domain.....“

We added in line 153 of the introduction:

„We describe here that AcuB is capable to bind and inhibit AcuC in presence of AMP to coordinate the activities of AcuA and
AcuC.“

An in lines 161-164:

„It was reported earlier that SrtN and AcuC are needed in *B. subtilis* for growth under conditions of low acetate³⁹. We show
here that AcuB allows to coordinate the activity of AcsA under conditions of high cellular acetate ensuring not two pathways
are active to generate acetyl-CoA from acetate.“

**Point 7:**

Please run basic spell/grammar checks on the entire MS. See for e.g pg 7 line 197 „..molar access..“, line 199 „...similar
extend.“, 209 „...analysed which role the presence...“. On Pg 9 line 270 „..BsAcuC with BsAcuC..“. Also I found
highlighted text which were remnants of revisions carried out by the authors.

**Response:**

We checked grammar and writing as suggested.

Reviewer #3 (Remarks to the Author):

Acetate metabolism is important to maintain the cellular energy balance in bacteria. The consumption and accumulation of acetate were precisely-regulated in bacteria to adapt the changed environment especially the available of carbon sources. In this work, the authors found that the AcuB, encoded by acu-operon, could reversibly bind AcuC in different cellular AMP level, thus resulting in inactivation or activation of AcuC. The AcuC is a deacetylase functioned in regulate the activity of AMP-forming acetyl-CoA-synthetase AcsA. This finding filled an important gap in regulation of AcsA activity and the coordination of acetate assimilation and dissimilation. The manuscript is well conceptualized and the data are presented in a logical manner. I have a few comments for the authors to address:

Point 1:

This work focused on the acu-operon in gram-positive Bacilli. Is acu-operon or acu-like operon existed in gram-negative bacteria such as *Escherichia coli*? Whether AcuB is a widespread energy sensor in regulating acetate assimilation and dissimilation in bacteria?

Response:

The acu-operon is only present in *Bacilli* and *Clostridia*, i.e., in *Firmicutes*. Moreover, to our knowledge there is no AcuB-like protein present in *E. coli*. However, in *E. coli* and even in humans the activity of the AMP-forming acetyl-CoA synthetase is regulated by lysine acetylation regulated by lysine acetyltransferases and lysine deacetylases. However, in *E. coli* there is no classical deacetylase present, making the presence of an AcuB-like protein obsolete, and AMP-forming acetyl-CoA synthetase is deacetylated and re-activated by the sirtuin deacetylase CobB. In humans, AMP-forming acetyl-CoA synthetase is deacetylated by SIRT3 in the mitochondria or SIRT1 in the nucleus. In eukaryotes, classical deacetylases are reported to exist in multi-protein complexes apart from some classical deacetylases including HDAC8.

Point 2:

From the aspect of biotechnology, is AcuB a potential bioengineering target for enhancing the acetate assimilation in bacteria?

Response:

This is an interesting question raised by the reviewer, which is difficult to answer. The regulation of acetate assimilation and dissimilation is tightly regulated at several layers. It is transcriptionally regulated by the CcpA, which is responsive to the cellular glucose level. Additionally, the acetate-kinase (AK)/phosphotransacetylase (Pta) pathway is active for acetate assimilation under conditions of high cellular acetate and depends on formation of acetyl-phosphate. Moreover, AcsA (AMP-forming acetyl-CoA synthetase) activity depends on its acetylation state, i.e. lysine acetyltransferases (KATs) depend on acetyl-CoA/CoA and sirtuins on NAD⁺ for acetylation and deacetylation of AcsA. This allows to adjust the activity of AcsA to the cellular metabolic state, i.e. acetylation in presence of high cellular acetyl-CoA inactivates AcsA activity and deacetylation by AcuC can restore its activity. Additionally, we showed recently that AcsA activity itself can be regulated by cellular acetyl-phosphate and CoA levels. From our data we conclude, that AcuB is important to coordinate acetate assimilation and dissimilation and ensures that not two parallel pathways are active under conditions of high cellular acetate that result in formation of acetyl-CoA (acetate assimilation), i.e. AK/Pta-pathway and AcsA. To this end, AcuB could be used to switch of AcsA activity under conditions of high cellular AMP-concentrations. Knocking-out AcuB might improve acetate assimilation under conditions of high cellular acetate, which would result in formation of AMP upon formation of acetyl-CoA by AcsA. However, when AcuC activity is not regulated by AcuB and cellular acetyl-CoA concentrations increase AcuA is able to acetylate and inactivate AcsA. This would open a futile cycle of acetylation of AcsA by AcuA and deacetylation by AcuC, which is only regulated by presence of acetyl-CoA needed for acetylation of AcsA by AcuA. From this discussion, we would assume that the sole knockout of AcuB would not result in a strong biotechnological benefit improving acetate assimilation if not done in a context of other mutations/deletions in strain optimization. In biotechnological processes with high nutrient availability we expect that cells are not lacking ATP and AcuB would not be active as inhibitor for AcuC activity under these conditions. However, there might be applications that could benefit from investigating the role of AcuB.

Point 3:

I suggest divide some long paragraphs into shorter ones.

Response:

We agree with reviewer 3 and subdivided the paragraph entitled „AcuB adopts different conformational states dependent
on the nucleotide-loading state“ into two paragraphs adding the new experimental data and AlphaFold3 models of AcuB
in complexes with various adenine nucleotides in the paragraph entitled „AcuB and AcuC form a heterotetrameric
AcuB₂•2AcuC-complex“ to improve readability.

**Point 4:**

I suggest reorganize Fig. 8 for better coordinating the regulation lines, energy flows and acetate metabolism pathways.

**Response:**

We reorganized the figure as suggested by the reviewer and prepared a new figure (Fig. 8d).

Point-by-point response to the reviewer's comments

Reviewer #1 (Remarks to the Author):

Response:

We thank the reviewer again for carefully reading our manuscript and for giving these constructive comments. We think that the manuscript did strongly improve due to these comments and by addressing these open questions in the revised manuscript.

Point 1:

Most of my previous comments have been addressed, except for points 1 and 8. Regarding the characterization of the nucleotide-dependent interaction between AcuB and AcuC, the authors have included additional data such as nanoDSF experiments and the Fluor de Lys assay. However, these newly added data do not directly measure protein-protein interaction. Rather, they reflect changes in protein conformation, not the actual binding affinity. Since a key conclusion of this manuscript is that AMP enhances the interaction between AcuB and AcuC, a quantitative method to assess the binding affinity between the two proteins would be preferable. Such an approach could directly demonstrate the extent to which different nucleotides modulate their interaction. If this is not feasible, the authors should explicitly discuss the limitations of the current study.

Response:

One open point is that the reviewer wants us to quantitatively measure the interaction of AcuC and AcuB and supply affinities for the interaction and the impact of the different nucleotides on this interaction. We have to admit that we do not directly measure the AcuB-AcuC interaction. We do measure, though, formation of a new larger species by gel filtration, which we cannot interpret other than formation of a complex. We also measure the effect on the enzymatic activity of AcuC, which only in the presence of both AcuB and AMP is strongly inhibited. Again, we cannot interpret this other than by formation of a complex. These data allow us to set an upper limit of the dissociation constant for the interaction of AcuC towards AcuB in presence of AMP. As already written in the first version of the manuscript we performed isothermal titration calorimetry (ITC) experiments with AcuB and AcuC with performing buffer exchange to buffers containing the respective adenine-nucleotides and also to assess the interaction without addition of nucleotides. However, these measurements were not successful due to the instability of the protein(s). As stated in the manuscript we faced problems with the stability of the AcuB-protein particularly in the nucleotide free state. Along that line, the fact that we always observe app. 25% of loading with nucleotide suggests that at least one adenine nucleotide binding site must be occupied within an AcuB dimer in order to increase its stability to prevent precipitation. To this end, we decided to perform measurements assessing the inhibitory potency of AcuB to inhibit the catalytic activity of AcuC dependent on the nucleotides present by a Fluor-de-Lys assay. These data allow us to directly assess the impact of AcuB binding on the catalytic activity of AcuC and thereby set an upper limit of the dissociation constant for the interaction between AcuB and AcuC depending on presence/absence of nucleotides. As shown by Fluor-de-Lys, immunoblotting and stability assays with nanoDSF, the presence of only nucleotides does neither interfere with AcuC catalytic activity nor stability. The measurement of the IC_{50} values was conducted by titrating AcuC with increasing concentrations of AcuB and using deacetylation of a Fluor-de-Lys substrate by AcuC as a readout (Revised manuscript: Fig. 7b). This resulted in an IC_{50} of 38 nM for the inhibitory potency of AcuB on AcuC activity in presence of AMP. Although not exactly quantifiable, for the other nucleotides the inhibitory potency of AcuB was strongly reduced (Fig. 7b). The fact that AcuC can be inhibited by AcuB in presence of AMP requires a direct interaction of AcuC and AcuB thereby blocking active site access of substrate. This is also suggested by the AlphaFold3 models and the mutant AcuB M189R incapable of binding and inhibiting AcuC. As a consequence, this furthermore suggests that the measured IC_{50} -value sets an upper limit for the affinity of AcuB towards AcuC (Cheng-Prusoff equation: $K_i = (IC_{50}/(1+[S]/K_M))$). In turn, this means that the affinity between AcuB and AcuC in presence of AMP is at least 38 nM or even stronger. For the other nucleotides we do not observe a similar inhibitory potency supporting our model that in presence of AMP AcuB has the highest potency to inhibit AcuC. From our perspective this clearly answers the remaining question addressed by the reviewer.

Apart from these data, we confirmed the interaction of AcuB and AcuC in presence of AMP by several independent assays, which all use different readouts. We performed analytical size-exclusion chromatography showing quantitative complex formation between AcuB and AcuC only in presence of AMP. Moreover, assessing deacetylation of its physiological substrate AMP-forming acetyl-CoA synthetase AcsA assessing its acetylation state by immunoblotting also shows that AcuB has the strongest inhibitory capacity in presence of AMP. Finally, assessing the activity of AcuC by deacetylation of a Fluor-de-Lys model substrate gave the same results. From our perspective there is no indication that the working model presented in this paper is not valid. This is the major finding of our study and the key message of our paper.

As a summary, we showed interaction of AcuC and AcuB in presence of AMP using different experimental methods and all led to the same result. Performing titrations with AcuB allows us to determine IC_{50} values indicating a strong interaction of AcuB towards AcuC in presence of AMP in the lower nanomolar range. This from our perspectives supports our hypothesis and also answers the open questions addressed by the reviewer.

**Point 2:**

In my original comments, I also noted that “the excessively long paragraph without breaks is unfriendly to readers.”
Unfortunately, this issue remains in the revised version. The Discussion section consists of a single paragraph spanning six
pages, and the Introduction similarly contains an unbroken paragraph that extends over four pages.

**Response:**

We agree with the reviewer and worked on all paragraphs of the manuscript and subdivided them into suitable subsections to
improve readability. Moreover, we shortened the introduction and discussion by removing repetitions and very general
information not essential for this study. In the current state, we think that the introduction and discussion contain only
information essential to understand the study.
